# Diffusion-based Layer-wise Semantic Reconstruction for Unsupervised Out-of-Distribution Detection

**Ying Yang**[1*], **De Cheng**[1*†], **Chaowei Fang**[1*†], **Yubiao Wang**[1]
**Changzhe Jiao**[1], **Lechao Cheng**[2], **Nannan Wang**[1], **Xinbo Gao**[3]
[1]Xidian University
[2]Hefei University of Technology
[3]Chongqing University of Posts and Telecommunications
{yycfq, ybwang_3}@stu.xidian.edu.cn
chenglc@hfut.edu.cn, gaoxb@cqupt.edu.cn
{dcheng, cwfang, cjiao, nnwang}@xidian.edu.cn

## Abstract

Unsupervised out-of-distribution (OOD) detection aims to identify out-of-domain data by learning only from unlabeled In-Distribution (ID) training samples, which is crucial for developing a safe real-world machine learning system. Current reconstruction-based method provides a good alternative approach, by measuring the reconstruction error between the input and its corresponding generative counterpart in the pixel/feature space. However, such generative methods face the key dilemma, *i.e.*, *improving the reconstruction power of the generative model, while keeping compact representation of the ID data.* To address this issue, we propose the diffusion-based layer-wise semantic reconstruction approach for unsupervised OOD detection. The innovation of our approach is that we leverage the diffusion model's intrinsic data reconstruction ability to distinguish ID samples from OOD samples in the latent feature space. Moreover, to set up a comprehensive and discriminative feature representation, we devise a multi-layer semantic feature extraction strategy. Through distorting the extracted features with Gaussian noises and applying the diffusion model for feature reconstruction, the separation of ID and OOD samples is implemented according to the reconstruction errors. Extensive experimental results on multiple benchmarks built upon various datasets demonstrate that our method achieves state-of-the-art performance in terms of detection accuracy and speed. Code is available at https://github.com/xbyym/DLSR.

## 1 Introduction

Unsupervised Out-of-Distribution (OOD) detection aims to identify whether a data point belongs to the In-Distribution (ID) or OOD dataset, by learning only from unlabeled in-distribution training samples. OOD detection plays a vital role in developing a safe real-world machine learning system, which ensures that the model is only performed on data drawn from the same distribution as its training data. If the test data does not follow the training distribution, the model could unintentionally produce nonsensical predictions, resulting in some misleading conclusions. Naturally, OOD detection is one of the key techniques for ensuring the model's robustness and safety.

Existing research studies the OOD detection mainly under two settings, *i.e.*, supervised and unsupervised. The supervised OOD detection methods usually deem this task as a binary classification problem, which relies on training with data labeled as OOD from disjoint categories or adversaries

---

* Equation Contribution. † Corresponding authors: De Cheng and Chaowei Fang.

38th Conference on Neural Information Processing Systems (NeurIPS 2024).

[Hendrycks et al., 2018], [Ming et al., 2022]. However, in many practical applications, it is almost impossible to access representative OOD samples, as the OOD data usually can be highly diverse and unpredictable. Therefore, we prefer to study the more challenging while practical unsupervised OOD detection problem. We will build an OOD detector trained solely on unlabeled ID data, as large amounts of unlabeled data are readily available and widely utilized due to their ease of acquisition.

Current reconstruction-based methods provide a good alternative approach for OOD detection, by measuring the reconstruction error between the input and its corresponding generative counterpart in the pixel/feature space. Obviously, the generative models and metric learning evaluation strategies are the main research directions. However, such methods of the generative models always face the following key dilemma: The projected in-distribution latent feature space should be compressed sufficiently to capture the exclusive characteristics of ID images, while it should also provide sufficient reconstruction power for the large-scale ID images of various categories. Existing generative-based methods (*e.g.*, auto-encoder (AE), variational AE [Kingma and Welling, 2013] and Generative Adversarial Network(GAN)) [Goodfellow et al., 2014], can not always fulfill these two requirements simultaneously, and a good balance between them is required. Besides, recent OOD detection methods based on diffusion models such as [Graham et al., 2023], [Gao et al., 2023] and [Liu et al., 2023] often involve the pixel-level reconstruction of distorted images, which consume much training/inference time and computation resources.

To address the above-mentioned issues, and inspired by the latent space noise addition mechanism in Latent Diffusion Models (LDM) Rombach et al. [2022], we propose the diffusion-based layer-wise semantic reconstruction approach for unsupervised OOD detection. Specifically, the proposed method makes full use of the diffusion model's intrinsic data reconstruction ability, to distinguish in-distribution samples from OOD samples in the latent feature space. In the diffusion denoising probabilistic models (DDPM) [Ho et al., 2020], the model is trained to incrementally remove noise from the noised inputs of different levels. Clearly, we can see that, instead of faithfully reconstructing inputs from the distribution it was trained on as previous VAE Kingma and Welling [2013] or GAN Goodfellow et al. [2014], the diffusion-based model shows more powerful reconstruction capabilities. Practically, our model involves reconstructing an input image feature from multiple values of the time step, this allows a single trained model to handle large amount of noise applied to the input, obviating the need for any dataset-specific fine-tuning.

Moreover, to set up a comprehensive and discriminative feature representation, we devise a multi-layer semantic feature extraction strategy. Performing feature reconstruction on top of the multi-layer semantic features, encourages to restrict the in-distribution latent features distributed more compactly within a certain space, so as to better rebuild in-distribution samples while not reconstructing OOD comparatively. Overall, by distorting the extracted multi-layer features with Gaussian noises and applying the diffusion model for feature reconstruction, the separation of ID and OOD samples is implemented according to the reconstruction errors. Note that, the proposed Latent Feature Diffusion Network (LFDN) is built on top of the feature level instead of the traditional pixel level, which could significantly improve the computation efficiency and achieve effective OOD detection. The other potential strength of such a strategy is that it avoids the reconstruction of minor characteristics unrelated to image understanding. In summary, the contributions of this work are as follows:

- We propose a diffusion-based layer-wise semantic reconstruction framework to tackle OOD detection, based on multi-layer semantic feature distortion and reconstruction. Meanwhile, We are the first to successfully incorporate generative modeling of features within the framework of OOD detection in image classification tasks.
- The layer-wise semantic feature reconstruction encourages restricting the in-distribution latent features to be more compactly distributed within a certain space, enabling better reconstruction of ID samples while limiting the reconstruction of OOD samples.
- Extensive experiments on multiple benchmarks across various datasets show that our method achieves state-of-the-art detection accuracy and speed.

## 2 Related Work

Existing researches study the OOD detection mainly under two settings: supervised and unsupervised. The Supervised method is generally based on classification. The method usually uses the maximum softmax probability [Hendrycks and Gimpel, 2016] from the final fully connected (FC) layer as the score to judge the ID sample. But the classification-based OOD detection methods often encounter

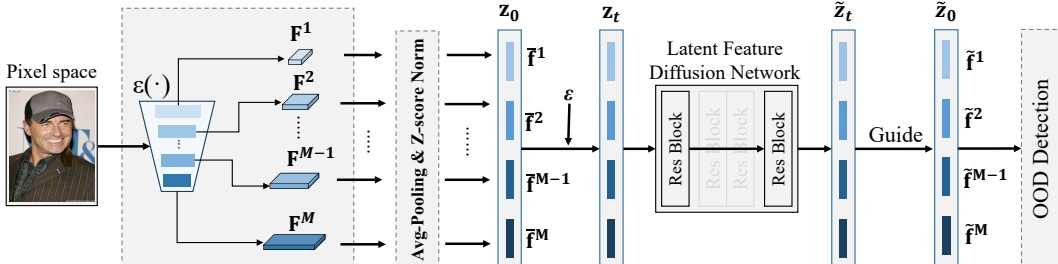

Figure 1: Overview of proposed diffusion-based layer-wise semantic reconstruction framework for unsupervised OOD detection. It includes multi-layer semantic feature extraction, Diffusion-based Feature Distortion and Reconstruction, and OOD detection head modules.

issues with assigning high softmax probability to OOD samples. Recent works [Liu et al., 2020], [Sun and Li, 2022], [Djurisic et al., 2022], [Zhao et al., 2024], attempt to alleviate this issue. The unsupervised OOD detection can be roughly categorized as the distance-based metric evaluation and the generative-based reconstruction methods.

Distance-based methods assume that OOD data lies far from ID class centroids. [Ren et al., 2021] improved OOD detection by separating image foregrounds from backgrounds and computing the Mahalanobis distance for each, then combining them. [Sun et al., 2022] used a non-parametric nearest neighbor distance for OOD detection. [Techapanurak et al., 2020] and [Chen et al., 2020] used cosine similarity to measure distances between test data features of in-distribution data to identify OOD data. [Huang et al., 2020] applied Euclidean distance, while [Gomes et al., 2022] used Geodesic distance for OOD detection. These methods often fail to capture sample distribution accurately.

Among the generative-based methods, the Likelihood-based methods can be traced back to as early as [Bishop, 1994]. This method assumes that the generative model assigns high likelihood to ID data, while the likelihood for OOD data tends to be lower. Recently, several deep generative models have supported the computation of likelihood, such as VAE [Kingma and Welling, 2013], PixelCNN++ [Salimans et al., 2017], and Glow [Kingma and Dhariwal, 2018]. However, some studies ([Nalisnick et al., 2018]; [Choi et al., 2018]; [Kirichenko et al., 2020]) have found that probabilistic generative models might also assign high likelihood to OOD data.

A series of studies have attempted to mitigate this issue. [Serrà et al., 2019] explored the relationship between image complexity and likelihood values, which adjusted likelihoods based on the size of image compression. [Ren et al., 2019] enhanced OOD detection by comparing likelihood values derived from different models. Another closely related approach highlights that these indicators are not well suited for VAEs. [Xiao et al., 2020] proposed a specialized metric known as likelihood regret for OOD detection in VAEs. [Cai and Li, 2023] suggested to leverage the high-frequency information of images to improve the model's ability to recognize OOD data. Additionally, a range of studies [Nalisnick et al., 2019], [Wang et al., 2020], [Bergamin et al., 2022], [Osada et al., 2023], have proposed typicality tests, estimating layer activation distributions and other statistical measures on training data, which are then evaluated through hypothesis testing or density estimation.

Another type of OOD detection methods leverage the idea that generative networks produce different reconstruction errors for ID and OOD data. Some methods such as [Sakurada and Yairi, 2014], [Zong et al., 2018], and [Zhou and Paffenroth, 2017], used auto-encoders to analyze reconstruction errors. GAN-based methods [Schlegl et al., 2017], [Zenati et al., 2018], and [Madzia-Madzou and Kuijf, 2022] utilized reconstruction errors and discriminator confidence to detect anomalies. Recent works [Graham et al., 2023], [Gao et al., 2023], and [Liu et al., 2023] applied diffusion models to model the pixel-level distribution of images, using errors from multiple reconstructions for OOD detection. Different from previous methods, we propose to leverage diffusion models to perform multi-layer semantic reconstruction in the latent feature space, not only for their stability in generation but also for significantly reducing training and inference time costs.

## 3 Method

Unsupervised OOD detection leverages intrinsic information from an unlabeled ID dataset $\mathbb{D}$ to train a detector. Suppose $\mathbb{D}$ contains $N$ images, namely $\mathbb{D} = \{\mathbf{x}_i\}_{i=1}^{N}$, where $\mathbf{x}_i$ denotes the $i$-th image.

The target is to learn an OOD detector denoted as $\mathcal{S}(\cdot)$, which can effectively evaluate an OOD score for each input image. The judgment of whether the input image belongs to ID or OOD is implemented by thresholding the OOD score. For example, given a testing image $\mathbf{x}$, it is recognized as an ID sample if the OOD score $\mathcal{S}(\mathbf{x})$ is lower than the pre-defined threshold $\lambda$; otherwise, it is recognized as an OOD sample.

In this paper, we propose a diffusion-based layer-wise semantic reconstruction framework to accomplish the OOD detection task. Specifically, as illustrated in Figure 1, the proposed framework consists of the following three components: the multi-layer semantic feature extraction module, the latent feature diffusion stage, and the OOD detection head.

## 3.1 Multi-layer Semantic Feature Extraction

The proposed semantic reconstruction-based method achieves OOD detection by measuring the reconstruction error between the input and its generative counterpart in the feature space. Specifically, we devise a multi-layer semantic feature extraction strategy, to set up a comprehensive and discriminative feature representation for each input image. Such multi-layer features could better rebuild the samples and encourage the ID semantic features distributed more compactly within a certain space from different semantic layers.

Specifically, given an image $\mathbf{x} \in \mathbb{R}^{3 \times w \times h}$ with $w$ and $h$ being the width and height of the input image, passing through an image encoder $\mathcal{E}(\cdot)$, (*e.g.*, EfficientNet [Tan and Le, 2019]), we can extract its feature maps from different layers (*i.e.*, low-level to high-level semantic blocks). The multi-layer intermediate feature map from the $m$-th block can be defined as $\mathbf{F}^m \in \mathbb{R}^{c_m \times w_m \times h_m}, m \in \{1, ..., M\}$, where $c_m$, $w_m$ and $h_m$ are the number of channels, width and height of the feature map $\mathbf{F}^m$, and $M$ is the total number of intermediate feature maps. Then, each feature map $\mathbf{F}^m$ undergoes the global average pooling, obtaining the one-dimensional feature vector $\mathbf{f}^m \in \mathbb{R}^{c_m}$. Afterward, Z-score normalization [Al Shalabi et al., 2006] is applied to each feature vector $\mathbf{f}^m$, resulting in $\bar{\mathbf{f}}^m = \frac{\mathbf{f}^m - \mu_{\mathbf{f}^m}}{\sqrt{\mathrm{Var}(\mathbf{f}^m) + \delta}}$ for the $m$-th intermediate feature vector $\mathbf{f}^m$ of the input image $\mathbf{x}$, where $\mathrm{Var}(\mathbf{f}^m)$ is the variance of $\mathbf{f}^m$ along the channel elements, and $\delta$ is a small constant value. Finally, we obtain the overall multi-layer feature vector for the input image $\mathbf{x}$ as: $\mathbf{z}_0 = \mathcal{H}(\mathbf{x}) = [\bar{\mathbf{f}}^1, \ldots, \bar{\mathbf{f}}^m, \ldots, \bar{\mathbf{f}}^M] \in \mathbb{R}^c$ by concatenating all the intermediate feature vectors, where $c = \sum_{m=1}^{M} c_m$, and $\mathcal{H}(\mathbf{x})$ denotes the whole feature extraction process.

## 3.2 Diffusion-based Feature Distortion and Reconstruction

Fitting the semantic feature distribution of ID samples is crucial for identifying whether the input is an ID or OOD sample. However, it is difficult to explicitly model the semantic feature space which has moderate complexity. Existing generative-based models [Zhou, 2022], [Cai and Li, 2023] address the modeling of complex data/feature space by transferring the original data/features into an implicit bottleneck space and learning a generator capable of recovering ID samples from the bottleneck space. Since the generator can not generalize well in recovering unseen OOD samples, it can be used as the OOD detector. Inspired by this, we set up a diffusion-based feature distortion and reconstruction framework, considering

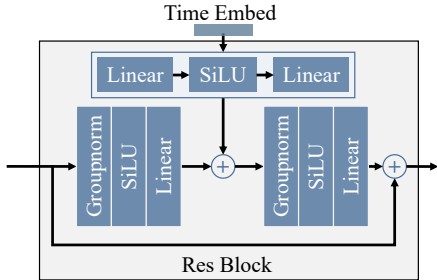

Figure 2: Residual Block Structure in LFDN.

the strength of diffusion models in data reconstruction. Our framework is innovative in the introduction of diffusion models in modeling semantic features, while previous works [Graham et al., 2023], [Liu et al., 2023], [Gao et al., 2023] focus on applying diffusion models for straightforward pixel-level distortion and reconstruction.

**Semantic Feature Distortion.**

The semantic feature distortion process can be conceptualized as transforming the semantic features into distorted counterparts with different levels of noise. For each step $t$ belonging to $[1, \ldots, T]$, the

generation of data point $\mathbf{z}_t$ follows the formula:

$$\mathbf{z}_t = \text{ennoise}(\mathbf{z}_0, t) = \sqrt{\overline{\alpha}_t} \times \mathbf{z}_0 + \sqrt{1 - \overline{\alpha}_t} \times \boldsymbol{\epsilon}, \quad \boldsymbol{\epsilon} \sim \mathcal{N}(\mathbf{0}^c, \mathbf{I}^{c \times c}) \tag{1}$$

where $\boldsymbol{\epsilon} \in \mathbb{R}^c$ represents a Gaussian noise vector; $\mathcal{N}(\cdot, \cdot)$ denotes the Gaussian distribution; $\mathbf{0}^c$ and $\mathbf{I}^{c \times c}$ denote the $c$-dimensional zero vector and the $c \times c$ identity matrix, respectively. $\overline{\alpha}_t$ is a predefined noise level that controls the amount of noise added at each step.

**Semantic Feature Reconstruction.** For reconstructing the semantic features from their distorted counterparts, we build up a Latent Feature Diffusion Network (LFDN) constituted by 16 residual blocks (ResBlock), as shown in Fig. 1.

The structure of ResBlock is illustrated in Fig. 2. Its residual branch is formed with two groups of Groupnorm [Wu and He, 2018], SiLU, and linear layers, as well as a MLP used for absorbing in the time embedding.

Following the calculation process of the denoising diffusion implicit model [Song et al., 2020], we employ LFDN to remove the noises injected into the semantic features with skipping step stride denoted as $s$. The detailed noise-removing process for $\mathbf{z}_t$ is described as follows. $s$ is set to a value randomly selected from $\{1, 2, \cdots, t\}$.

1) We first input $\mathbf{z}_t$ and the time embedding of $t$ into LFDN, generating an initial reconstruction state denoted as $\tilde{\mathbf{z}}_t$. The calculation formulation can be summarized as: $\tilde{\mathbf{z}}_t = \text{LFDN}(\mathbf{z}_t, t)$, where $\text{LFDN}(\cdot)$ denotes the feed-forward process of LFDN.

2) Afterwards, we estimate the noise correction vector for $\mathbf{z}_t$ denoted as $\tilde{\boldsymbol{\epsilon}}_t$ as follows,

$$\tilde{\boldsymbol{\epsilon}}_t = \frac{\mathbf{z}_t - \sqrt{\overline{\alpha}_t} \times \tilde{\mathbf{z}}_t}{\sqrt{1 - \overline{\alpha}_t}}, \tag{2}$$

where $\overline{\alpha}_t$ is the predefined noise level of the $t$-th feature distortion step.

3) Then, we sample the input ($\tilde{\mathbf{z}}_{t'}$) for implementing the $t'$-th step's feature reconstruction where $t' = \max(t - s, 0)$ as:

$$\tilde{\mathbf{z}}_{t'} = \sqrt{\overline{\alpha}_{t'}} \left( \frac{\mathbf{z}_t - \sqrt{1 - \overline{\alpha}_t} \times \tilde{\boldsymbol{\epsilon}}_t}{\sqrt{\overline{\alpha}_t}} + \sqrt{1 - \overline{\alpha}_{t'} - \sigma_t^2} \times \tilde{\boldsymbol{\epsilon}}_t \right) + \sigma_t^2 \boldsymbol{\epsilon}, \tag{3}$$

where $\sigma_t^2$ represents the variance of the additional noise at step $t$. Regarding $\tilde{\mathbf{z}}_{t'}$ and time embedding of $t'$ as inputs, LFDN predicts reconstruction results of the $t'$-th step as $\tilde{\mathbf{z}}_{t'} = \text{LFDN}(\tilde{\mathbf{z}}_{t'}, t')$.

4) Repeating steps 2 and 3 until $t' = 0$, yields the final reconstructed semantic features $\tilde{\mathbf{z}}_0$.

We summarize the above process as $\tilde{\mathbf{z}}_0 = \text{denoise}(\mathbf{z}_t, t)$. This framework ensures that $\tilde{\mathbf{z}}_0$ is not solely derived from the LFDN output but is continuously refined by DDIM, integrating detailed corrections to achieve high accuracy in reconstructing the original data from its noisy observations.

**Objective Function**. For optimizing the network parameters of LFDN, the mean square error is used as the loss function for pulling close the outputs of LFDN with the original semantic features. The calculation formulation is as follows:

$$L = \frac{1}{N} \sum_{\mathbf{x} \in \mathbb{D}} \|\mathbf{z}_0 - \text{LFDN}(\mathbf{z}_t, t)\|_2^2 \tag{4}$$

During training, $t$ is randomly selected from $\{1, 2, \cdots, T\}$. The detail is illustrated in Algorithm 1.

## 3.3 OOD Detection Head

Our approach can be integrated with three metrics to detect OOD data. Firstly, we utilize the Mean Squared Error (MSE) to measure the feature reconstruction error. Secondly, we use the Likelihood Regret metric (LR = MSE$_{\text{initial}}$ − MSE$_{\text{final}}$) [Xiao et al., 2020], which quantifies the change in MSE from the initial epoch to the final epoch. This metric reflects the model's evolving certainty during training. Generally, the reconstruction errors for ID samples decrease as the model becomes more familiar with these samples, whereas the errors for OOD samples remain relatively stable.

Lastly, we employ the Multi-layer Semantic Feature Similarity (MFsim), $i.e.$, the cosine similarity. We assesses the cosine similarity between the original features $\mathbf{z}_0 = [\bar{\mathbf{f}}^1, \ldots, \bar{\mathbf{f}}^m, \ldots, \bar{\mathbf{f}}^M]$ and the reconstructed features $\tilde{\mathbf{z}}_0 = [\tilde{\mathbf{f}}^1, \ldots, \tilde{\mathbf{f}}^m, \ldots, \tilde{\mathbf{f}}^M]$ at various layers: $\text{Sim}(\bar{\mathbf{f}}^m, \tilde{\mathbf{f}}^m) = \frac{\bar{\mathbf{f}}^m \cdot \tilde{\mathbf{f}}^m}{\|\bar{\mathbf{f}}^m\| \cdot \|\tilde{\mathbf{f}}^m\|}$. The OOD detection score MFsim, is then computed as the negative average of these similarities: $\text{MFsim} = -\frac{1}{M}\sum_{m=1}^{M}\text{Sim}(\bar{\mathbf{f}}^m, \tilde{\mathbf{f}}^m)$, where $M$ is the number of feature maps. A higher MFsim score indicates a greater likelihood of the data being OOD. Algorithm 2 details the MFsim calculation. The flows for MSE and LR calculations are provided in Appendix A.

---

**Algorithm 1** Training Algorithm

1: **Input:** Train image $\mathbf{x} \in \mathbb{R}^{3 \times h \times w}$
2: $\mathbf{z}_0 = \mathcal{H}(\mathbf{x}) = [\bar{\mathbf{f}}^1, \ldots, \bar{\mathbf{f}}^m, \ldots, \bar{\mathbf{f}}^M] \in \mathbb{R}^c$
3: **repeat**
4:     Draw $t \sim \text{Uniform}\{1, \ldots, T\}$
5:     Draw $\epsilon \sim \mathcal{N}(0, I)$
6:     Compute $\mathbf{z}_t$ and $L$
7:     $\mathbf{z}_t = \sqrt{\bar{\alpha}_t}\mathbf{z}_0 + \sqrt{1 - \bar{\alpha}_t}\epsilon$
8:     $L = \frac{1}{N}\sum_{\mathbf{x} \in \mathbb{D}}\|\mathbf{z}_0 - \text{LFDN}(\mathbf{z}_t, t)\|_2^2$
9:     Update the parameters via the AdamW optimizer.
10: **until** convergence

---

**Algorithm 2** Testing Algorithm

1: **Input:** An image $\mathbf{x} \in \mathbb{R}^{3 \times h \times w}$
2: **Output:** OOD score
3: $\mathbf{z}_0 = \mathcal{H}(\mathbf{x}) = [\bar{\mathbf{f}}^1, \ldots, \bar{\mathbf{f}}^m, \ldots, \bar{\mathbf{f}}^M] \in \mathbb{R}^c$
4: $\mathbf{z}_t \leftarrow \text{ennoise}(\mathbf{z}_0, t)$
5: $\tilde{\mathbf{z}}_0 \leftarrow \text{denoise}(\mathbf{z}_t, t)$
6: $[\tilde{\mathbf{f}}^1, \ldots, \tilde{\mathbf{f}}^m, \ldots, \tilde{\mathbf{f}}^M] \leftarrow \tilde{\mathbf{z}}_0$
7: **for** $m = 1$ to $M$ **do**
8:     $S_m \leftarrow \text{Sim}(\bar{\mathbf{f}}^m, \tilde{\mathbf{f}}^m)$
9: **end for**
10: $\text{MFsim} \leftarrow -\left(\sum_{m=1}^{M}S_m\right)/M$
11: **return** MFsim

---

# 4 Experiments

## 4.1 Datasets and Evaluation Metrics

**Datasets**: We train the OOD detection model on three in-distribution (ID) datasets: CIFAR-10 [Krizhevsky et al., 2009], CIFAR-100, and CelebA [Liu et al., 2015]. When testing models learned on a specific ID dataset, we select several datasets from SVHN [Netzer et al., 2011], SUN [Xiao et al., 2010], LSUN-c [Yu et al., 2015], LSUN-r, iSUN [Xu et al., 2015], iNaturalist [Van Horn et al., 2018], Textures [Cimpoi et al., 2014], Places365 [Zhou et al., 2017], MNIST [Deng, 2012], FMNIST, KMNIST [Clanuwat et al., 2018], Omniglot [Lake et al., 2015], and NotMNIST as OOD data.

**Evaluation Metrics**: We employed the area under the receiver operating characteristic (AUROC) and the false positive rate at 95% true positive rate (FPR95) as evaluation metrics. Results in FPR95 metric are provided in Appendix C.1.

## 4.2 Implementation Details

We utilize EfficientNet-b4 [Tan and Le, 2019] or ResNet50 [He et al., 2016] pre-trained on ImageNet [Deng et al., 2009] as our encoder. The main text presents results using EfficientNet-b4, while results using ResNet50 are detailed in Appendix C.2. For EfficientNet-b4, we select feature maps from the first to fifth stages ($M = 5$) to construct the multi-layer semantic features, resulting in a feature dimension ($c$) of 720. The LFDN is consisting of 16 residual blocks. Inside each residual block, the number of groups in Groupnorm and the intermediate feature dimension of the residual branch are set to 1 and 1440, respectively. We employ the AdamW optimizer with a weight decay of $10^{-4}$. Our method is trained on NVIDIA Geforce 4090 GPU for 150 epochs, with a batch size of 128 and a constant learning rate of $10^{-4}$ throughout the training phase.

## 4.3 Comparison with State-of-the-art Methods

**Compared Generative-based Methods:** In Table 1, regarding CIFAR-10 as the ID dataset, we compare our method against pixel-level generative-based methods including GLOW [Serrà et al., 2019], PixelCNN++ [Serrà et al., 2019], VAE [Xiao et al., 2020], and DDPM [Graham et al., 2023]. To validate the effectiveness of LFDN, we implement a variant of our method through replacing LFDN with AutoEncoder in which MFsim is used for estimating the OOD score. In comparison with the best

pixel-level method, VAE, our method achieves a 9.1% improvement in average AUROC when using MFsim for OOD score estimation. Compared to DDPM, our method variants show a significantly improvement in average AUROC. For example, when integrated with MSE, our method achieves 20.4% higher AUROC than DDPM. This indirectly indicates that performing OOD detection at the pixel level is much worse than performing OOD detection at the feature level. Generating pixels may reconstruct more content unrelated to the image's semantics, which may interfere the identification of OOD samples. Making the model focus on the reconstruction of compactly distributed semantic features benefits in separating ID and OOD samples. In terms of testing speed, our method is nearly 100 times faster than DDPM, significantly enhancing performance while reducing detection costs. Moreover, the final version of our method built upon LFDN improves average AUROC by 18.5% compared to the variant basd on AutoEncoder, as the diffusion model captures data distribution more effectively.

In Table 2, we compare our method with VAE, DDPM and AutoEncoder, using CelebA as the ID dataset. Our method integrated with MFsim achieves state-of-the-art performances, with an AUROC improvement of 19.89% compared to DDPM, and the performance of the remaining two metrics also far exceeds the baseline, demonstrating the generalizability of our approach.

**Compared Classification-based and Distance-based Methods:** In Table 3, we compare our method with classification-based methods including MSP [Hendrycks and Gimpel, 2016], EBO [Liu et al., 2020], DICE [Sun and Li, 2022], and ASH-S [Djurisic et al., 2022], as well as distance-based methods including 'SimCLR+Mahalanobis Distance' [Xiao et al., 2021] and 'SimCLR+KNN' [Sun et al., 2022]. All methods are evaluated using EfficientNet-b4 as the backbone. Compared to classification-based and distance-based methods, our approach consistently shows a clear advantage. Specifically, for CIFAR-100 as the in-distribution dataset, our method integrated with MFsim achieves an average AUROC of 13.84% higher than the classification-based method DICE. Moreover, unlike classification-based methods, our approach does not require labeled data.

The inference speed of our method based on MSE or MFsim is faster than that of distance-based methods SimCLR+Maha and SimCLR+KNN, because the computation of covariance matrix or K nearest neighbors occupies part of time. Our method is also comparable to classifier-based methods including MSP, EBO, DICE and ASH-S. This demonstrates the effectiveness of leveraging the strong ability of diffusion models to reconstruct original distributions from different noise levels for reconstructing low-dimensional features and performing OOD detection.

Table 1: The AUROC values for OOD detection, where CIFAR-10 is used as the in-distribution dataset. The results are compared with generative-based methods. Higher AUROC values indicate better performance, with the best results highlighted in bold for clarity.

| Dataset | | Pixel-Generative-Base | | | | Feature-Generative-Base | | | |
|---|---|---|---|---|---|---|---|---|---|
| ID | OOD | GLOW | PixelCNN++ | VAE | DDPM | AutoEncoder | our(+MSE) | ours(+LR) | ours(+MFsim) |
| CIFRA10 | SVHN | 88.3 | 73.7 | 95.9 | 97.3 | 57.7 | 97.3±0.0 | 98.2±0.0 | **98.9±0.1** |
| | LSUN | 21.3 | 64.0 | 40.3 | 68.2 | 81.5 | 97.6±0.1 | 97.8±0.1 | **99.8±0.1** |
| | MNIST | 85.8 | 96.7 | **99.9** | 83.2 | 95.8 | 99.4±0.0 | 98.9±0.1 | **99.9±0.0** |
| | FMNIST | 71.2 | 90.7 | 99.1 | 84.3 | 79.6 | 99.0±0.0 | 98.8±0.0 | **99.9±0.0** |
| | KMNIST | 38.0 | 82.6 | **99.9** | 89.7 | 90.5 | 99.5±0.0 | 99.1±0.0 | **99.9±0.0** |
| | Omniglot | 95.5 | 98.9 | 99.6 | 35.9 | 81.5 | 99.1±0.1 | 97.1±0.1 | **99.9±0.0** |
| | NotMNIST | 53.9 | 82.6 | 99.4 | 88.7 | 81.6 | 99.8±0.1 | 99.5±0.0 | **99.9±0.0** |
| | average | 64.9 | 84.2 | 90.6 | 78.2 | 81.2 | 98.8±0.1 | 98.5±0.1 | **99.7±0.1** |
| Time | Num img/s (↑) | 38.6 | 19.3 | 0.7 | 11.4 | 1224.2 | 999.3 | 273.6 | 999.3 |

## 4.4 Ablation Study

**Illustration of the generation ability of the diffusion model on OOD detection**. To demonstrate the evolution of the generative model's reconstruction capability for both ID and OOD samples before and after training, we compare the distributions of the MFsim scores at the first epoch and the final epoch in **Figure 3**. CIFAR-10 serves as the ID dataset, while the other six datasets listed in **Table 3** are employed as OOD data. Our observations reveal that the diffusion model's reconstruction ability enhances across most datasets, with a notably more pronounced improvement for the in-distribution samples. This indicates that ID samples are reconstructed more effectively, thereby validating the efficacy of our method.

Table 2: The AUROC values for OOD detection, where CelebA is used as the in-distribution dataset. The results are compared with generative-based methods. Higher AUROC values indicate better performance, with the best results highlighted in bold for clarity.

| Dataset | | Pixel-Generative-Based | | Feature-Generative-Based | | | |
|---|---|---|---|---|---|---|---|
| ID | OOD | VAE | DDPM | AutoEncoder | ours(+MSE) | ours(+LR) | ours(+MFsim) |
| CelebA | SUN | 95.89 | 83.41 | 32.90 | **99.98±0.01** | 97.15±0.02 | **99.98±0.01** |
| | iNaturalist | 95.52 | 82.38 | 41.56 | **100+0.00** | 99.96±0.01 | 99.99±0.00 |
| | Textures | 91.73 | 78.33 | 56.33 | 99.93±0.02 | 98.51±0.02 | **99.96±0.01** |
| | Places365 | 97.58 | 76.25 | 35.90 | 99.96±0.01 | 97.47±0.03 | **99.98±0.00** |
| | average | 95.18 | 80.09 | 41.67 | 99.97±0.01 | 98.27±0.02 | **99.98±0.01** |
| Time | Num img/s (↑) | 18.7 | 10.2 | 1357.6 | 1033.8 | 290.4 | 1033.8 |

Table 3: The AUROC values for OOD detection, where CIFAR-10/100 is used as the in-distribution dataset. The results are compared with Classification-based and Distance-based methods using EfficientNet-b4 as the backbone. Higher AUROC values indicate better performance, with the best results highlighted in bold for clarity.

| ID | Based | Method | Num img/s (↑) | OOD | | | | | | average |
|---|---|---|---|---|---|---|---|---|---|---|
| | | | | SVHN | LSUN-c | LSUN-r | iSUN | Textures | Places365 | |
| CIFAR10 | Classifier-based | MSP | 1060.5 | 94.53 | 96.37 | 91.80 | 92.23 | 95.93 | 97.59 | 94.74 |
| | | EBO | 1060.5 | 96.79 | 97.34 | 94.42 | 94.64 | 96.30 | 98.34 | 96.31 |
| | | DICE | 1066.3 | 98.53 | 99.03 | 94.49 | 95.25 | 97.68 | 99.63 | 97.44 |
| | | ASH-S | 1047.6 | 98.01 | 98.23 | 93.17 | 94.13 | 97.01 | 98.48 | 96.51 |
| | Distance-based | SimCLR+Mahalanobis | 674.8 | 97.80 | 73.61 | 69.28 | 88.63 | 76.47 | 67.42 | 78.87 |
| | | SimCLR+KNN | 919.8 | 92.40 | 92.05 | 89.81 | 90.14 | 97.24 | 94.36 | 92.67 |
| | Generative-based | ours(+MSE) | 960.6 | 97.31±0.02 | 97.59±0.01 | 93.93±0.01 | 92.78±0.01 | 100±0.00 | 99.96±0.00 | 96.93±0.01 |
| | | ours(+LR) | 360.2 | 98.22±0.02 | 97.84±0.02 | 95.37±0.01 | 94.31±0.02 | 100±0.00 | 99.91±0.01 | 97.61±0.02 |
| | | ours(+MFsim) | 960.6 | **98.89±0.01** | **99.83±0.02** | **98.83±0.01** | **98.52±0.02** | 100±0.00 | 100±0.00 | **99.34±0.01** |
| CIFAR100 | Classifier-based | MSP | 1060.5 | 77.56 | 84.03 | 72.09 | 71.52 | 90.02 | 89.00 | 80.70 |
| | | EBO | 1060.5 | 76.51 | 81.59 | 78.92 | 76.38 | 79.38 | 83.07 | 79.31 |
| | | DICE | 1066.3 | 86.93 | 88.54 | 71.97 | 71.29 | 92.83 | 90.78 | 83.72 |
| | | ASH-S | 1047.6 | 92.11 | 90.03 | 63.30 | 65.12 | 95.25 | 92.99 | 83.13 |
| | Distance-based | SimCLR+Mahalanobis | 674.8 | 56.24 | 52.23 | 61.34 | 73.53 | 71.92 | 51.98 | 61.21 |
| | | SimCLR+KNN | 919.8 | 54.37 | 51.49 | 83.80 | 77.21 | 53.31 | 54.43 | 62.44 |
| | Generative-based | ours(+MSE) | 960.6 | 83.93±0.01 | 86.86±0.01 | 75.38±0.01 | 71.99±0.02 | 99.99±0.00 | 99.97±0.01 | 86.35±0.01 |
| | | ours(+LR) | 360.2 | 88.84±0.01 | 87.60±0.02 | 80.96±0.01 | 77.71±0.02 | 99.98±0.01 | 99.92±0.02 | 89.17±0.01 |
| | | ours(+MFsim) | 960.6 | **93.90±0.01** | **99.14±0.01** | **95.74±0.01** | **94.40±0.01** | 100±0.00 | 100±0.00 | **97.20±0.01** |

**Performance variations across different sampling time steps**: **Figure 4** illustrates the variations in average AUROC and FPR95 values for different evaluation metrics at various sampling time steps, using CIFAR-10 as the ID data, with the final time step $T = 100$. It is observed that all metrics perform poorly at $t = 1$ primarily due to minimal noise added, making $\mathbf{z}_t$ too similar to $\mathbf{z}_0$ and thus, limiting the denoising capability of LFDN; both ID and OOD data are well reconstructed. As $t$ increases to about 3-10 steps, the appropriate amount of noise allows MSE, LR, and MFsim to reach optimal performances. However, as $t$ continues to increase, the difference between $\mathbf{z}_t$ and the original $\mathbf{z}_0$ enlarges, with $\mathbf{z}_t$ gradually approaching random noise, thereby worsening the reconstruction differences between $\tilde{\mathbf{z}}_0$ and $\mathbf{z}_0$ for both ID and OOD samples.

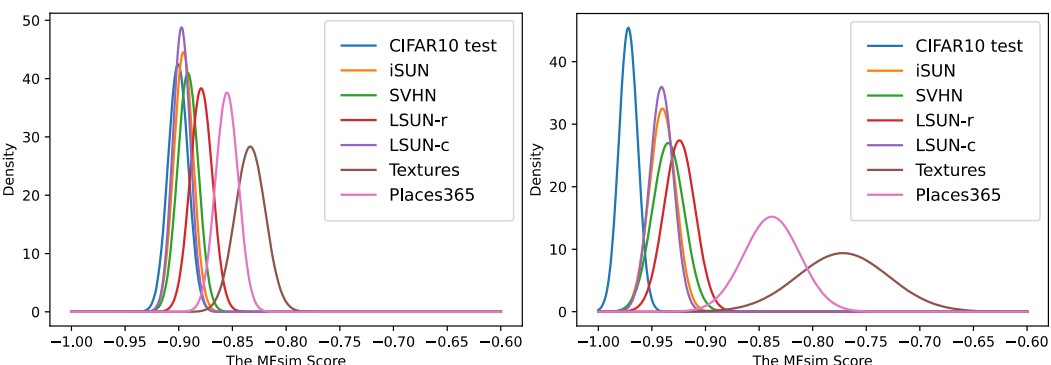

Figure 3: The MFsim score distributions of the first epoch (left) and the last epoch (right)

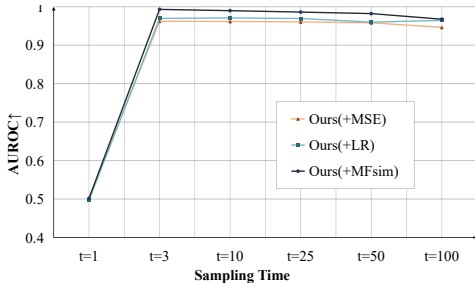 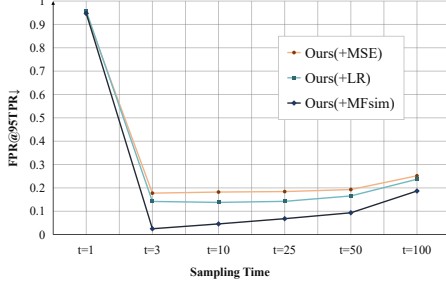

Figure 4: CIFAR-10 dataset is the ID data, the six datasets listed in Table 3 are used as OOD data. The average AUROC and FPR95 for the three metrics are evaluated at different sampling time steps.

Table 4: Changes in Average AUROC Across Six Datasets listed in Table 3 for CIFAR100 as ID.

| Metrics | MSE | | LR | | MFsim | |
|---|---|---|---|---|---|---|
| Linear | Linear=720 | Linear=1440 | Linear=720 | Linear=1440 | Linear=720 | Linear=1440 |
| **Average** | 83.35 | 86.35 | 84.05 | 89.17 | 96.43 | 97.20 |
| Number of Blocks | Number=8 | Number=16 | Number=8 | Number=16 | Number=8 | Number=16 |
| **Average** | 85.26 | 86.35 | 87.32 | 89.17 | 97.13 | 97.20 |

**Comparison of MFsim across different feature scales**. **Figure 5** displays performance comparisons of MFsim when reconstructing the last block (i.e., $f_4, C = 448$) versus multi-layer semantic features under an EfficientNet-b4 encoder. The results demonstrate that multi-layer semantic features generally outperform single-layer ones, indicating that multi-layer semantic features contain richer semantic information and are more representative of samples across different in-distribution datasets. Furthermore, considering the diverse semantic information represented by different layers, combing various layers of semantic features helps to boost the OOD performances of LFDN.

**Ablation study on LFDN network parameters**. We conducted ablation experiments on two groups of parameters within the LFDN network: the dimension of the linear layers and the number of ResBlocks. For each experiment, we reduced one of these parameters to half of its original size while keeping all other parameters unchanged. **Table 4** presents the results of these experiments, showing how these modifications affect the performance. It is observed that the performance of our MFsim metric remains relatively stable, indicating that it continues to provide effective OOD detection capabilities even under conditions of reduced network size.

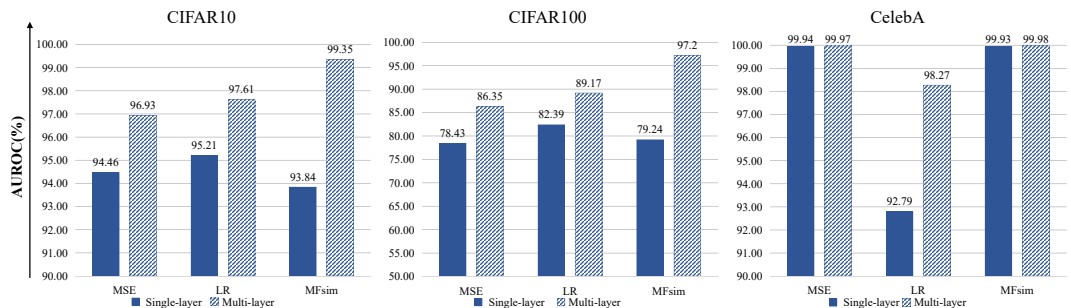

Figure 5: Variation of Average AUROC Values across Different Scales

# 5 Conclusion and Limitation

In this paper, we propose a diffusion-based layer-wise semantic reconstruction framework for unsupervised OOD detection. We leverage the diffusion model's intrinsic data reconstruction ability to distinguish in-distribution and OOD samples in the latent feature space. Specially, the diffusion-based feature generation is built on top of the devised multi-layer semantic feature extraction strategy, which

sets up a comprehensive and discriminative feature representation benefiting the generative OOD detection methods. Finally, we hope our proposed OOD detection method could make contributions to develop a safe real-world machine learning system. Additionally, it needs to point out that the performance of our method also relies on the quality of features extracted by the encoder. Therefore, selecting an encoder with strong feature extraction capabilities is crucial for achieving good performances.

## 6 Acknowledgement

This work was supported in part by the National Key R&D Program of China under Grant No.2023YFA1008600, in part by NSFC under Grant NO.62376206, 62176198 and U22A2096, in part by the Key R&D Program of Shaanxi Province under Grant 2024GX-YBXM-135, in part by the Key Laboratory of Big Data Intelligent Computing under Grant BDIC-2023-A-004.

## 7 Broader Impacts

Positive Societal Impacts: The proposed diffusion-based layer-wise semantic reconstruction method for unsupervised out-of-distribution (OOD) detection can significantly enhance the security and safety of machine learning systems. By effectively identifying OOD data, the system can prevent incorrect or potentially harmful decisions, making AI applications more reliable in critical areas such as healthcare, autonomous driving, and financial systems. This method increases the robustness of AI systems by ensuring they can handle unexpected inputs gracefully. This contributes to the overall stability and trustworthiness of AI deployments in various industries, thereby promoting wider acceptance and integration of AI technologies. Negative Societal Impacts: As with any advanced detection method, there is a risk that the technology could be misused. For instance, surveillance applications, it could be employed to monitor individuals without their consent, leading to privacy violations and ethical concerns.

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

# Appendix

## A  Supplementary Algorithm

This section provides two key algorithms used for evaluating our approach: the MSE (Mean Squared Error) calculation and the Likelihood Regret (LR) calculation.

The MSE calculation, as shown in Algorithm 3, computes the mean squared error between the original and reconstructed latent features. It serves as a basic measure of reconstruction error for detecting OOD samples.

The LR calculation, detailed in Algorithm 4, measures the reduction in reconstruction error by comparing the MSE values at the initial and final epochs of training. This metric reflects how well the model has adapted to the ID data over time, with a higher reduction indicating better adaptation.

---
**Algorithm 3** Testing Algorithm for MSE Calculation

---
1: **Input:** An image $\mathbf{x}$
2: **Output:** MSE score
3: $\mathbf{z}_0 = \mathcal{H}(\mathbf{x})$
4: $\mathbf{z}_t \leftarrow \text{ennoise}(\mathbf{z}_0)$
5: $\tilde{\mathbf{z}}_0 = \text{denoise}(\mathbf{z}_t, t)$
6: $\text{MSE} \leftarrow \frac{1}{N} \sum_{i=1}^{N} (\mathbf{z}_0[i] - \tilde{\mathbf{z}}_0[i])^2$          $\triangleright$ $i$ indexes the elements of $\mathbf{z}_0$ and $\tilde{\mathbf{z}}_0$
7: **return** MSE

---

---
**Algorithm 4** Testing Algorithm for LR Calculation

---
1: **Input:** An image $\mathbf{x}$, MSE at initial and final epochs
2: **Output:** LR score
3: $\mathbf{z}_0^{\text{initial}} = \mathcal{H}(\mathbf{x})$ at the beginning of training
4: $\mathbf{z}_t^{\text{initial}} \leftarrow \text{ennoise}(\mathbf{z}_0^{\text{initial}})$
5: $\tilde{\mathbf{z}}_0^{\text{initial}} = \text{denoise}(\mathbf{z}_t^{\text{initial}}, t)$
6: $\text{MSE}_{\text{initial}} \leftarrow \frac{1}{N} \sum_{i=1}^{N} (\mathbf{z}_0^{\text{initial}}[i] - \tilde{\mathbf{z}}_0^{\text{initial}}[i])^2$
7: $\mathbf{z}_0^{\text{final}} = \mathcal{H}(\mathbf{x})$ at the end of training
8: $\mathbf{z}_t^{\text{final}} \leftarrow \text{ennoise}(\mathbf{z}_0^{\text{final}})$
9: $\tilde{\mathbf{z}}_0^{\text{final}} = \text{denoise}(\mathbf{z}_t^{\text{final}}, t)$
10: $\text{MSE}_{\text{final}} \leftarrow \frac{1}{N} \sum_{i=1}^{N} (\mathbf{z}_0^{\text{final}}[i] - \tilde{\mathbf{z}}_0^{\text{final}}[i])^2$
11: $\text{LR} \leftarrow \text{MSE}_{\text{initial}} - \text{MSE}_{\text{final}}$
12: **return** LR

---

## B  More Experimental Details

### B.1  Dataset Details and Testing Speeds

**Table 1 : CIFAR-10 Dataset**    The CIFAR-10 test set consisted of 10,000 images. The SVHN dataset contained 26,032 images, LSUN-r had 10,000 images, and Fashion-MNIST, MNIST, and KMNIST each comprised 10,000 images. Omniglot included 13,180 images, and notMNIST had 18,724 images, totaling 97,936 OOD samples. The testing of the MFsim metric took a total of 98 seconds, with an average speed of 999.3 images per second.

**Table 2 : CelebA Dataset**    The CelebA test set comprised 60,780 images, SUN included 10,000 images, iNaturalist had 100,000 images, Textures consisted of 1,678 images, and Places365 had 1,002 images, making up a total of 112,680 OOD samples. Testing the MFsim metric took a total of 109 seconds, processing an average of 1033.8 images per second.

## B.2 Training Details

Both CIFAR-10 and CelebA datasets were trained for 200 epochs using the VAE model. The GLOW model was trained for 150 epochs with a learning rate of $5 \times 10^{-4}$, and PixelCNN+ was trained for 150 epochs at the same learning rate. Under the DDPM model, both datasets were trained for 350 epochs, following the experimental setups and code provided in the original papers. We used LFDN without time-step embeddings as our autoencoder, used MFsim metrics, and kept all remaining training details consistent with our approach.

# C Supplementary Experiments

## C.1 Experimental Results for FPR95 Values Using EfficientNet-b4 as Backbone

We conducted tests to evaluate the FPR95 (False Positive Rate at 95% True Positive Rate) values using CIFAR10 and CIFAR100 datasets as in-distribution data while treating the remaining six datasets as out-of-distribution datasets. The specific FPR95 values are summarized in **Table 5**.

Table 5: The FPR95 values for OOD detection, where CIFAR-10/100 is used as the in-distribution dataset. The results are compared with Classification-based and Distance-based methods using EfficientNet-b4 as the backbone. Higher AUROC values indicate better performance, with the best results highlighted in bold for clarity.

| ID | Based | Method | Num img/s (↑) | OOD | | | | | | average |
|---|---|---|---|---|---|---|---|---|---|---|
| | | | | SVHN | LSUN-c | LSUN-r | iSUN | Textures | Places365 | |
| CIFAR10 | Classifier-based | MSP | 1060.5 | 43.99 | 26.13 | 48.65 | 46.89 | 27.50 | 15.03 | 34.70 |
| | | EBO | 1060.5 | 16.51 | 11.52 | 28.38 | 27.03 | 16.08 | 4.99 | 17.42 |
| | | DICE | 1066.3 | 7.70 | 4.81 | 25.74 | 21.76 | 7.80 | 1.49 | 11.55 |
| | | ASH-S | 1047.6 | 6.89 | 4.15 | 31.29 | 26.29 | 5.21 | 1.32 | 12.53 |
| | Distance-based | SimCLR+Mahalanobis | 674.8 | 9.24 | 67.73 | 75.43 | 64.32 | 56.22 | 72.15 | 57.52 |
| | | SimCLR+KNN | 919.8 | 49.15 | 54.89 | 76.97 | 73.48 | 15.27 | 39.39 | 51.53 |
| | Generative-based | ours(+MSE) | 960.6 | 21.15±0.03 | 19.52±0.01 | 39.67±0.02 | 43.76±0.02 | **0±0.00** | 0.42±0.03 | 20.75±0.02 |
| | | ours(+LR) | 360.2 | 14.26±0.02 | 18.67±0.03 | 31.62±0.02 | 37.76±0.02 | 0.06±0.01 | 0.83±0.02 | 17.20±0.02 |
| | | ours(+MFsim) | 960.6 | **4.34±0.02** | **0.04±0.01** | **4.42±0.02** | **6.26±0.02** | **0±0.00** | **0±0.00** | **2.51±0.01** |
| CIFAR100 | Classifier-based | MSP | 1060.5 | 80.10 | 68.80 | 80.35 | 80.36 | 47.11 | 57.41 | 69.02 |
| | | EBO | 1060.5 | 88.74 | 78.64 | 72.35 | 76.57 | 95.83 | 94.04 | 84.36 |
| | | DICE | 1066.3 | 63.77 | 58.96 | 77.89 | 78.67 | 34.26 | 48.77 | 60.39 |
| | | ASH-S | 1047.6 | **34.28** | 44.39 | 89.45 | 86.13 | 21.44 | 34.59 | 51.71 |
| | Distance-based | SimCLR+Mahalanobis | 674.8 | 94.95 | 96.35 | 85.05 | 86.29 | 80.37 | 95.50 | 89.75 |
| | | SimCLR+KNN | 919.8 | 95.32 | 97.11 | 78.45 | 84.38 | 95.28 | 94.82 | 90.89 |
| | Generative-based | ours(+MSE) | 960.6 | 89.05±0.02 | 69.14±0.03 | 69.63±0.02 | 89.79±0.01 | 0.12±0.02 | **0±0.00** | 52.95±0.02 |
| | | ours(+LR) | 360.2 | 62.06±0.03 | 72.19±0.04 | 86.67±0.02 | 86.17±0.02 | 0.42±0.02 | 2.81±0.02 | 51.72±0.03 |
| | | ours(+MFsim) | 960.6 | 37.48±0.02 | **1.90±0.01** | **23.05±0.02** | **26.00±0.02** | **0±0.00** | **0±0.00** | **14.78±0.02** |

As shown in **Table 5**, our method demonstrates a significant advantage in terms of FPR95 values compared to other classification-based and distance-based approaches. Specifically, when using CIFAR100 as in-distribution data, our method achieves an average reduction of 36.93% in FPR95 values compared to the state-of-the-art classification-based approach, ASH-S.

## C.2 Experimental Results with ResNet50 as Encoder

Besides using EfficientNet-b4 as the encoder, we also employed the commonly used ResNet50 to extract multi-layer semantic features. For ResNet50, feature maps from stages 1 to 3 were selected, with channel counts of 256, 512, and 1024, respectively. These feature maps were concatenated to form a 1792-dimensional vector, which was then used as input for the LFDN. The results for three OOD detection metrics are presented in **Table 6** and **Table 7**. Both tables compare our method with classification-based and distance-based methods.

As shown in **Table 6** and **Table 7**, when using ResNet50 as the backbone, our method still achieves the best performance. Specifically, with CIFAR-10 as the in-distribution dataset, the average AUROC and MFsim values are 98.30% and 8.89%, respectively, outperforming the classification-based SOTA method DICE by 6.17% in AUROC and reducing the FPR95 by 24.43%.

Figures 6 and 7 illustrate the differences in the MFsim score distributions for various datasets, with ResNet50 as the encoder and CIFAR10 as the in-distribution dataset, across the first and last epochs.

Table 6: The AUROC values for OOD detection, where CIFAR-10/100 is used as the in-distribution dataset. The results are compared with Classification-based and Distance-based methods using ResNet50 as the backbone. Higher AUROC values indicate better performance, with the best results highlighted in bold for clarity.

| ID | Based | Method | Num img/s (↑) | SVHN | LSUN-c | LSUN-r | iSUN | Textures | Places365 | average |
|---|---|---|---|---|---|---|---|---|---|---|
| | | | | | | OOD | | | | |
| CIFAR10 | Classifier-based | MSP | 1321.5 | 80.64 | 95.05 | 91.65 | 89.95 | 87.78 | 90.43 | 89.25 |
| | | EBO | 1321.5 | 81.90 | 98.21 | 94.43 | 93.09 | 86.84 | 92.45 | 91.15 |
| | | DICE | 1369.3 | 91.92 | 99.18 | 91.13 | 90.51 | 87.39 | 92.66 | 92.13 |
| | | ASH-S | 1307.4 | 84.16 | 98.76 | 95.00 | 94.40 | 87.88 | 91.63 | 91.97 |
| | Distance-based | SimCLR+Mahalanobis | 857.2 | 90.17 | 73.74 | 86.14 | 83.25 | 81.48 | 91.23 | 84.34 |
| | | SimCLR+KNN | 1179.8 | 92.36 | 91.13 | 87.78 | 91.82 | 88.62 | 79.41 | 88.52 |
| | Generative-based | ours(+MSE) | 654.9 | 81.40 | 94.02 | 81.11 | 81.36 | 100.00 | 100.00 | 89.65 |
| | | ours(+LR) | 296.7 | 90.95 | 92.53 | 85.22 | 85.91 | 100.00 | 100.00 | 92.44 |
| | | ours(+MFsim) | 654.9 | **95.98±0.02** | **99.86±0.02** | **97.06±0.02** | **96.89±0.01** | **100±0.00** | **100±0.00** | **98.30±0.01** |
| CIFAR100 | Classifier-based | MSP | 1321.5 | 78.38 | 84.18 | 78.98 | 78.09 | 76.54 | 72.00 | 78.03 |
| | | EBO | 1321.5 | 83.13 | 89.35 | 83.83 | 83.35 | 78.85 | 71.42 | 81.66 |
| | | DICE | 1369.3 | 87.93 | 93.32 | 82.41 | 82.20 | 79.29 | 69.65 | 82.47 |
| | | ASH-S | 1307.4 | 91.66 | 93.24 | 69.93 | 72.78 | 87.75 | 71.00 | 81.06 |
| | Distance-based | SimCLR+Mahalanobis | 857.2 | **91.92** | 57.14 | 87.47 | 88.00 | 94.96 | 71.86 | 81.89 |
| | | SimCLR+KNN | 1179.8 | 87.78 | 84.30 | 82.51 | 77.69 | 83.35 | 73.74 | 81.56 |
| | Generative-based | ours(+MSE) | 654.9 | 86.55 | 99.11 | 93.02 | 91.87 | 100.00 | 100.00 | 95.09 |
| | | ours(+LR) | 296.7 | 89.17 | 99.16 | **93.65** | 92.33 | 100.00 | 100.00 | 95.72 |
| | | ours(+MFsim) | 654.9 | 89.68±0.02 | **99.18±0.01** | 93.64±0.02 | **92.94±0.02** | **100±0.00** | **100±0.00** | **95.91±0.01** |

Table 7: The FPR95 values for OOD detection, where CIFAR-10/100 is used as the in-distribution dataset. The results are compared with Classification-based and Distance-based methods using ResNet50 as the backbone. Higher AUROC values indicate better performance, with the best results highlighted in bold for clarity.

| ID | Based | Method | Num img/s (↑) | SVHN | LSUN-c | LSUN-r | iSUN | Textures | Places365 | average |
|---|---|---|---|---|---|---|---|---|---|---|
| | | | | | | OOD | | | | |
| CIFAR10 | Classifier-based | MSP | 1321.5 | 60.56 | 16.09 | 28.21 | 35.63 | 45.4 | 32.93 | 36.47 |
| | | EBO | 1321.5 | 59.15 | 8.07 | 23.79 | 30.72 | 60.2 | 31.66 | 35.60 |
| | | DICE | 1369.3 | 27.07 | 4.11 | 37.83 | 41.12 | 57.48 | 32.32 | 33.32 |
| | | ASH-S | 1307.4 | 53.93 | 5.57 | 20.29 | 22.31 | 64.36 | 39.71 | 34.36 |
| | Distance-based | SimCLR+Mahalanobis | 857.2 | 27.65 | 33.35 | 48.17 | 51.22 | 38.12 | 60.43 | 43.16 |
| | | SimCLR+KNN | 1179.8 | 24.53 | 25.29 | 37.81 | 27.55 | 34.36 | 62.19 | 35.29 |
| | Generative-based | ours(+MSE) | 654.9 | 94.79 | 45.99 | 95.86 | 91.68 | 0.00 | 0.00 | 54.72 |
| | | ours(+LR) | 296.7 | 34.59 | 25.77 | 56.54 | 52.05 | 0.00 | 0.00 | 28.16 |
| | | ours(+MFsim) | 654.9 | **21.10±0.03** | **0.02±0.01** | **15.75±0.03** | **16.48±0.04** | **0.00±0.00** | **0.00±0.00** | **8.89±0.02** |
| CIFAR100 | Classifier-based | MSP | 1321.5 | 53.38 | 43.52 | 56.23 | 57.69 | 63.63 | 77.53 | 58.66 |
| | | EBO | 1321.5 | 47.04 | 34.15 | 51.14 | 52.36 | 63.05 | 80.95 | 54.78 |
| | | DICE | 1369.3 | 38.7 | 28.77 | 56.21 | 56.74 | 65.21 | 82.63 | 54.71 |
| | | ASH-S | 1307.4 | **29.83** | 28.75 | 89.48 | 85.22 | 51.8 | 81.48 | 61.09 |
| | Distance-based | SimCLR+Mahalanobis | 857.2 | 32.19 | 80.43 | 39.93 | 40.39 | 28.21 | 81.44 | 50.43 |
| | | SimCLR+KNN | 1179.8 | 39.23 | 48.99 | 60.21 | 74.99 | 57.15 | 80.74 | 60.22 |
| | Generative-based | ours(+MSE) | 654.9 | 78.54 | **0.49** | 41.81 | 43.12 | 0.00 | 0.00 | 27.33 |
| | | ours(+LR) | 296.7 | 65.93 | 0.57 | **36.71** | **38.78** | 0.00 | 0.00 | **23.67** |
| | | ours(+MFsim) | 654.9 | 64.72±0.03 | 1.08±0.02 | 39.66±0.04 | 39.79±0.03 | **0.00±0.00** | **0.00±0.00** | 24.21±0.02 |

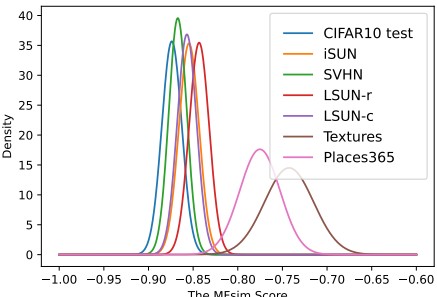

Figure 6: The MFsim score distributions of the First Epoch with ResNet50 as Encoder

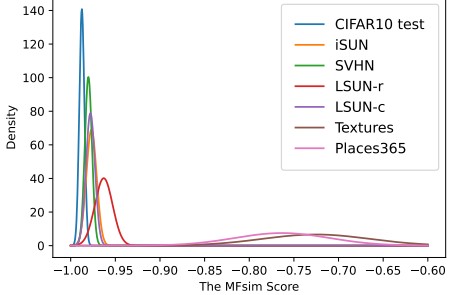

Figure 7: The MFsim score distributions of the Last Epoch with ResNet50 as Encoder

## C.3 CIFAR-10 as ID and CIFAR-100 as OOD

As shown in **Table 8**, when CIFAR-10 is used as the ID dataset and CIFAR-100 as the OOD dataset, our method consistently achieves the best performance across different evaluation metrics, including AUROC and FPR95. Compared to classification-based methods, the improvement is not significantly large, but our approach still shows a consistent edge, particularly in feature-generative-based models, demonstrating the robustness of our method.

Table 8: The FPR95 and AUROC Values for CIFAR-10 as ID Samples and CIFAR100 as OOD Samples.

| ID | OOD | Based | Method | FPR95↓ | AUROC↑ |
|---|---|---|---|---|---|
| | | | MSP | 52.04 | 86.14 |
| | | Classification-Based | EBO | 51.32 | 86.19 |
| | | | ASH-S | 51.29 | 87.13 |
| | | | GLOW | - | 73.60 |
| CIFAR10 | CIFAR100 | Pixel-Generative-Based | VAE | 90.41 | 55.95 |
| | | | DDPM | 67.38 | 82.43 |
| | | | ours(+MSE) | **48.87** | **87.54** |
| | | Feature-Generative-Based | ours(+LR) | 49.48 | 87.24 |
| | | | ours(+MFsim) | 53.70 | 85.60 |

## C.4 Comparisons with recent generative methods

The comparison between our method and DDPM[Graham et al., 2023] can be referred to **Table 1** and **Table 2**. Our method outperforms DDPM consistently on benchmarks using CIFAR10 or CelebA as ID data.

The comparison between our method and Diffuard[Gao et al., 2023] is provided in **Table 9**. Results of Diffuard are taken from its original paper. Here, CIFAR10 is regarded as ID data, while CIFAR100 or TinyImagenet is regarded as OOD data. Our method based on MFsim achieves overall better performance than 'Diffuard+Deep Ens', with 1.55 higher AUROC and 21.77 lower FPR95.

The comparison between our method and LMD[Liu et al., 2023] is shown in **Table 10**. The evaluation metric is AUROC. The average AUROC of our method based on MFsim is 6.94 higher than that of LMD.

Table 9: The AUROC and FPR95 values compared to DiffGuard [Gao et al., 2023] using CIFAR-10 as the ID dataset and CIFAR-100/TinyImageNet as the OOD datasets.

| Method | CIFAR-100 | | TINYIMAGENET | | average | |
|---|---|---|---|---|---|---|
| | AUROC↑ | FPR95↓ | AUROC↑ | FPR95↓ | AUROC↑ | FPR95↓ |
| Diffuard | 89.88 | 52.67 | 91.88 | 45.48 | 90.88 | 49.08 |
| Diffuard+EBO | 89.93 | 50.77 | 91.95 | 43.58 | 90.94 | 47.18 |
| Diffuaed+Deep Ens | **90.40** | 52.51 | 91.98 | 45.04 | 91.19 | 48.78 |
| ours(+MSE) | 87.54 | **48.87** | 97.68 | 13.42 | 92.61 | 31.15 |
| ours(+LR) | 87.24 | 49.48 | 97.11 | 15.04 | 92.18 | 32.26 |
| ours(+MFsim) | 85.60 | 53.70 | **99.88** | **0.39** | **92.74** | **27.01** |

## C.5 Comparison against other methods using the multi-scale feature encodings as the input.

In **Table 11**, we have made comparison of our method against AE and VAE using the multi-layer feature encodings as inputs. For AE (AutoEncoder), we use the LFDN network without the timestep embedding, i.e., a 16-layer linear network. For VAE, we use a 5-layer linear network as the encoder

Table 10: The AUROC values compared to LMD [Liu et al., 2023] using CIFAR-10/CIFAR-100 as the ID dataset and CIFAR-100/CIFAR10/SVHN as the OOD datasets.

| ID | OOD | LMD | ours(+MSE) | ours(+LR) | ours(+MFsim) |
|---|---|---|---|---|---|
| CIFAR10 | CIFAR100 | 60.70 | 87.54 | 87.24 | 85.6 |
| | SVHN | 99.20 | 97.31 | 98.22 | 98.89 |
| CIFAR100 | CIFAR10 | 56.80 | 70.52 | 72.86 | 64.58 |
| | SVHN | 98.50 | 83.93 | 88.84 | 93.9 |
| AVERAGE | | 78.80 | 84.83 | **86.79** | 85.74 |

and an 8-layer linear network as the decoder. Compared to AE and VAE, the diffusion model has significant advantages when modeling complex multidimensional distributions.

## C.6 Comparisons with pixel-level denoising approaches.

We provide the distribution differences of the MSE score and MFsim score at two levels after training, with CIFAR-10 as ID dataset and other datasets as OOD; The results are shown in Figures 8 and 9.

It can be observed that at the pixel level(DDPM), the reconstruction error distributions of ID and OOD samples are very similar. The mixed MSE scores make it very hard to distinguish ID samples from OOD samples. However, at the feature level, the reconstruction score distribution of ID samples shows a clear distinction from that of OOD samples. The reason is that, our feature-level diffusion-based generative model makes the projected in-distribution latent space not only be compressed sufficiently to capture the exclusive characteristics of ID images, but also provide sufficient reconstruction power for the large-scale ID images of various categories. In other words, the pretrained encoder has inherent generalization capabilities, and the multi-layer features it extracts are more discriminative than the high-dimensional pixels of the images themselves.

Table 11: The AUROC values compared to the other generative models using CIFAR-10 as the ID dataset.

| Dataset | | Method | | |
|---|---|---|---|---|
| ID | OOD | AE(+MFsim) | VAE(+MFsim) | Diffusion(+MFsim) |
| CIFRA10 | SVHN | 57.68 | 83.96 | 98.89 |
| | LSUN | 81.47 | 97.69 | 99.83 |
| | MNIST | 95.85 | 99.98 | 99.99 |
| | FMNIST | 79.61 | 98.69 | 99.99 |
| | KMNIST | 90.51 | 99.96 | 99.99 |
| | Omniglot | 81.50 | 97.69 | 99.99 |
| | NotMNIST | 81.61 | 99.88 | 99.99 |
| | average | 81.18 | 96.84 | **99.81** |
| Time | Num img/s (↑) | 1224.2 | 1179.4 | 999.3 |

## C.7 ImageNet100 as ID Dataset

In **Table 12**, our method using MSE outperforms the classification-based SOTA method DICE, achieving an improvement of 3.91% in AUROC when ImageNet100 is used as the ID dataset and various datasets such as SUN, iNaturalist, Textures, and Places365 are used as OOD datasets. The significant improvements in performance metrics demonstrate that our generative-based approach can effectively model the in-distribution characteristics, leading to better OOD detection capabilities. This indicates that our proposed method is particularly suitable for more complex datasets like ImageNet100, where capturing detailed features is crucial for accurate OOD detection.

Table 12: The AUROC and FPR95 Values for Different Methods with ImageNet100 as ID Dataset and SUN/iNaturalist/Textures/Places365 as OOD Datasets.

| ID | Method | # img/s (↑) | SUN | | iNaturalist | | Textures | | Places365 | | Average | |
|---|---|---|---|---|---|---|---|---|---|---|---|---|
| | | | AUROC↑ | FPR95↓ | AUROC↑ | FPR95↓ | AUROC↑ | FPR95↓ | AUROC↑ | FPR95↓ | AUROC↑ | FPR95↓ |
| ImageNet100 | MSP | 956.2 | 89.68 | 52.16 | 86.44 | 57.88 | 86.48 | 55.18 | 88.85 | 53.79 | 87.86 | 54.75 |
| | EBO | 956.2 | 90.83 | 49.46 | 87.82 | 58.29 | 86.04 | 66.81 | 90.01 | 51.80 | 88.68 | 56.59 |
| | DICE | 950.8 | 93.78 | 37.14 | **89.76** | **50.47** | 89.52 | 52.32 | 93.18 | 41.42 | 91.56 | 45.34 |
| | ASH-S | 977.5 | 90.12 | 44.54 | 86.94 | 51.45 | 88.38 | 45.23 | 88.76 | 49.70 | 88.55 | 47.73 |
| | ours(+MSE) | 954.9 | **98.81** | **5.91** | 86.59 | 71.88 | 97.68 | 10.76 | **98.79** | **5.73** | **95.47** | **23.57** |
| | ours(+LR) | 313.4 | 93.98 | 16.51 | 51.39 | 87.99 | 76.02 | 47.54 | 94.95 | 16.25 | 79.09 | 42.07 |
| | ours(+MFsim) | 954.9 | 98.49 | 8.41 | 84.63 | 77.86 | **98.08** | **9.98** | 98.37 | 8.75 | 94.89 | 26.25 |

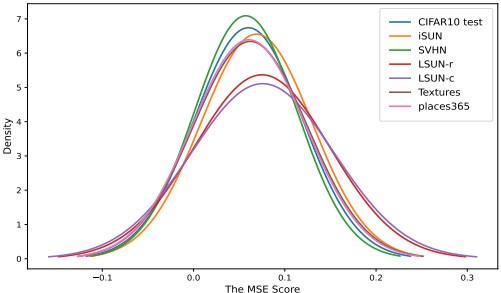

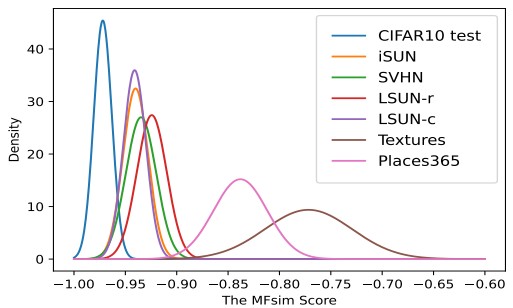

Figure 8: Reconstruction Error Distribution of ID and OOD Samples for Pixel-level

Figure 9: Reconstruction Error Distribution of ID and OOD Samples for Feature-level

# D    Qualitative results.

We have included three types of failure cases Figures 10, 11 and 12 . The first type, shown in Figure 10, represents ID samples misclassified as OOD. It can be observed that these misclassified samples often have significant shadows and lack semantic information, resulting in high reconstruction errors and being incorrectly classified as OOD samples. The second type, shown in Figure 11, represents OOD samples misclassified as ID. It can be observed that these OOD samples have categories very similar to those of the ID samples (CIFAR-10), such as cars and ships, which are categories present in CIFAR-10. The third type, shown in Figure 12, represents OOD samples with colors very similar to the ID samples, leading to their misclassification as ID.

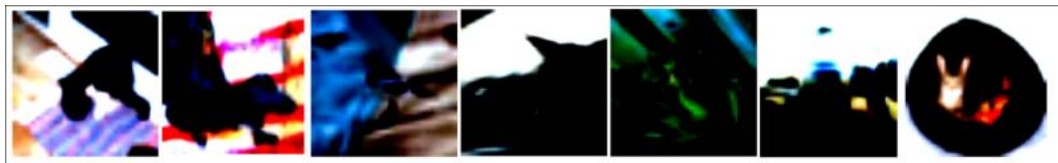

Figure 10: Examples of ID Samples Misclassified as OOD (Lacking Semantic Information).

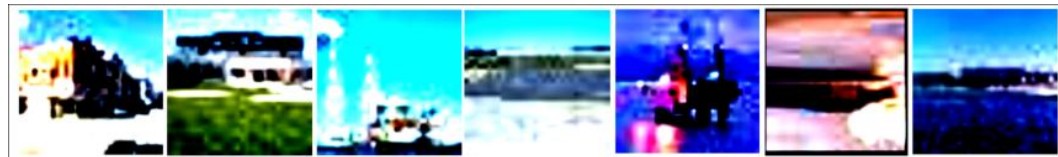

Figure 11: Examples of OOD Samples Misclassified as ID (Similar to ID Sample Categories).

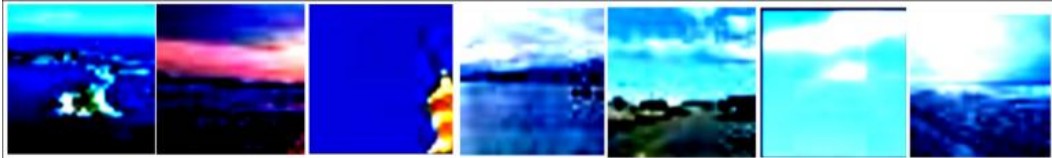

Figure 12: Examples of OOD Samples Misclassified as ID (Similar to ID Sample Colors).

