# OpenReview forum: "Diffusion-based  Layer-wise Semantic Reconstruction for Unsupervised Out-of-Distribution Detection"
_NeurIPS.cc/2024/Conference — NeurIPS 2024 poster_

### Official Review · Reviewer_Jgzd · 2024-06-13

**Soundness:** 2
**Presentation:** 3
**Contribution:** 2
**Rating:** 5
**Confidence:** 4

**Summary:**

This paper focuses on out of distribution detection. The detection is based on the reconstruction error. The authors use a diffusion model as their generative model and focus on the feature space instead of the original images. The authors test their proposed methods on several different datasets.

**Strengths:**

1. The overall presentation of this paper is good. It is very easy to follow.
2. Have conducted experiments on different datasets.

**Weaknesses:**

There are several concerns for this paper.

1) The novelty of this paper is very limited. It looks like it simply replaces a generative model with the diffusion model.

2) The current version only presents the "what"---for example, the model includes three components (multi-layer semantic feature extraction, diffusion model for feature reconstruction, and OOD detection head)---but not the "why". More explanations and theoretical analysis may need.

3) The evaluation metrics may be problematic. Due to the natural of the OOD problem, the dataset would be highly imbalanced. In addition to the AUROC and FPR95, other metrics that could capture the imbalance characteristics should be used.

4) The current ID and OOD are very different (from different datasets), what would happen if the difference between ID and OOD is not so significant? For example, both ID and OOD come from the same dataset (e.g., CIFAR10) but OOD is curated by adding some distortions.

**Questions:**

Basically, my major concerns include: 1) why it works? 2) the experiments. Please see [Weaknesses] for more details.

**Limitations:**

yes.

---

> ### Author Rebuttal · Authors · 2024-08-06
>
> 1. Novelty of this paper
>
> Thanks for the valuable comments. I believe the reviewer may misunderstand our contributions on OOD detection by stating that “It looks like it simply replaces a generative model with diffusion model”. Our main novelty lies in the following three aspects:
>
> Firstly, we are the first to successfully incorporate generative modeling of features within the framework of OOD detection in image classification tasks. This demonstrates that when using diffusion models for OOD detection in such downstream tasks, it is not necessarily required to operate in the pixel space.
>
> Secondly, we devise a multi-layer semantic feature reconstruction mechanism.  Performing feature reconstruction on top of the multi-layer semantic features, encourages to restrict the in-distribution latent features distributed more compactly within a certain space, so as to better rebuild in-distribution samples while not reconstructing OOD comparatively. As a result, the projected in-distribution latent feature space should be compressed sufficiently to capture the exclusive characteristics of ID images, while it also provides sufficient reconstruction power for the large-scale ID images of various categories with the high-level semantic features.
>
> Thirdly, the proposed Latent Feature Diffusion Network (LFDN) is built on top of the feature level instead of the traditional pixel level, which could significantly improve the computation efficiency and achieve effective OOD detection.
>
> 2. More explanations and theoretical analysis may need.
>
> Thanks for the valuable comments. We will improve these descriptions in the camera-ready version.
> As illustrated in Figure 1, the proposed framework consists of the following three components:
>
> (1)The multi-layer semantic feature extraction module, which sets up a comprehensive and discriminative feature representation for each image, and could help to better rebuild the samples and encourage the in-distribution features distributed more compactly within a certain space from different semantic layers.
>
> (2) The latent feature diffusion stage, which introduces the DDPM to model the multi-layer semantic features. It builds a latent feature diffusion network to reconstruct the semantic features from their distorted counterparts.
>
>  (3) The OOD detection head, which utilizes different evaluation metrics (i.e., MSE, MFsim and LR metric), to measure the reconstruction error. Finally, we can use the reconstruction error to justify whether the input sample belongs to ID or OOD.
>
> The success of our method lies in that, the projected in-distribution latent feature space can be compressed sufficiently to capture the exclusive characteristics of ID images, while it also provides sufficient reconstruction power for the large-scale ID images of various categories.
>
> Besides, to further illustrate the effectiveness of the proposed method, **we also compare the MFsim score distributions between the initial model and the final model after training. We can clearly see that the reconstruction errors of the ID data tend to decreasing as the model training, and the final scores can be distinguished from the OOD data.**
>
> 3. In addition to the AUROC and FPR95, other metrics that could capture the imbalance characteristics should be used.
>
> Besides the commonly used AUROC and FPR95 evaluation metrics, we also adopt the F1-score and AUPRC evaluation metrics which could capture the imbalance characteristics of the dataset, for comprehensive experimental comparison. As shown in the table below, we compare the proposed method with some of the latest representative Classification-based methods under the F1-score and AUPRC evaluation metrics, respectively.
>
> Comparison of F1-Score
>
> |ID|Method|SVHN|LSUN-c|LSUN-r|iSUN|Textures|Places365|**average**|
> |-|-|-|-|-|-|-|-|-|
> |CIFAR10|MSP|48.20|80.88|75.74|75.85|81.75|75.90|73.05|
> ||EBO|46.76|92.72|81.93|81.30|82.53|81.17|77.74|
> ||ASH-S|67.44|93.55|90.99|91.33|83.12|82.39|84.80|
> ||ours(+MSE)|79.21|91.04|83.16|82.93|97.43|97.43|88.53|
> ||ours(+LR)|86.18|91.85|85.74|85.03|97.43|97.43|90.61|
> ||ours(+MFsim)|91.91|97.42|95.20|94.71|97.43|97.43|**95.68**|
> |CIFAR100|MSP|47.92|72.47|70.29|72.08|78.51|68.02|68.22|
> ||EBO|46.81|75.34|70.91|72.39|78.48|67.84|68.63|
> ||ASH-S|44.35|74.83|67.94|69.86|78.20|68.79|67.33|
> ||ours(+MSE)|49.80|73.75|67.56|70.11|97.43|97.43|76.01|
> ||ours(+LR)|52.39|74.05|68.75|71.13|97.43|97.42|76.86|
> ||ours(+MFsim)|65.48|96.52|87.39|87.20|97.43|97.43|**88.58**|
>
> We provide the average AUPRC values across six different datasets in Table I of the PDF attached  in the global rebuttal.
>
> We can clearly see that our method also obtains the best performances under all the settings.
>
> Besides, to analyze the effect of the imbalanced dataset, we design different ratios of ID and OOD data samples for evaluations. As shown in the table below, the detection accuracy tends to decreasing as the sample ratio (ID: OOD) gets larger.
>
> |ID|ID:OOD Ratios|**average F1-Score**|
> |-|-|-|
> |CIFAR10|1:0.1|97.39|
> ||1:0.5|97.29|
> ||1:1|97.23|
> ||1:5.7|95.68|
> |CIFAR100|1:0.1|97.25|
> ||1:0.5|96.47|
> ||1:1|95.49|
> ||1:5.7|88.58|
>
>
> 4. What would happen if the difference between ID and OOD is not so significant?
>
> To analyze the effect of the difference between ID and OOD on the model performance, we also artificially design the following two experiments:
>
> 1) The ID dataset is from CIFAR10, and the OOD dataset is CIFAR100 which shares similar distributions with CIFAR10.
>
> The results are in Table III of the PDF.
>
> 2) Both the ID and OOD datasets are CIFAR10, while the OOD dataset is CIFAR10 by adding  random Gaussian noise.
>
> The results are as follows:
>
> |ID|OOD|Method|FPR95↓|AUROC↑|
> |-|-|-|-|-|
> |CIFAR10|CIFAR10(add noise)|MSP|90.23|61.63|
> |||EBO|88.01|62.49|
> |||ASH-S|86.95|62.16|
> |||VAE|89.20|51.76|
> |||DDPM|95.80|45.00|
> |||ours(+MSE)|7.84|98.43|
> |||ours(+LR)|16.27|96.76|
> |||ours(+MFsim)|**0.14**|**99.97**|
>
> We can clearly see that our method is superior to other methods.

---

> > ### Comment · Reviewer_Jgzd · 2024-08-13
> >
> > Thanks so much for your time on the rebuttal. I have carefully read it. Based on the rebuttal and other available reviews, I think most of my previous concerns have been addressed. I have raised my score accordingly.

---

> ### Author Response · Authors · 2024-08-10
> **Concerns addressed**
>
> Dear Reviewer Jgzd,
>
> Thank you very much for reviewing our paper and providing valuable feedback. We have made the best effort to address all your questions and comments. In particular, we have further clarified the rationale behind our proposed method and provided additional points to highlight its novelty. We have also introduced several metrics that can capture the characteristics of imbalanced datasets. Additionally, we demonstrated the detection performance of our method in scenarios where the ID and OOD datasets are similar, showcasing the scalability of our approach.
>
> We sincerely hope that our responses can address all your concerns. Is there anything that needs us to further clarify for the given concerns?
>
> Thank you again for your hard work and thoughtful review.

---

> ### Author Response · Authors · 2024-08-12
>
> Dear Reviewer Jgzd,
>
> Thank you very much for reviewing our paper and providing valuable feedback. We sincerely hope that our responses can address all the remaining concerns. Thank you again for your great help and many good questions and suggestions, which largely help improve the quality of our paper. We would like to clarify if you have further concerns.
>
> We would like to clarify if you have further concerns. Thanks very much.

---

### Official Review · Reviewer_aPqn · 2024-07-05

**Soundness:** 2
**Presentation:** 3
**Contribution:** 3
**Rating:** 6
**Confidence:** 4

**Summary:**

The authors introduce a method for unsupervised out-of-distribution (OoD) detection for image classification tasks. Their approach is based on the semantic reconstruction of latent features in multiple layers of an image encoder using a diffusion model and does not require any OoD data for training. Unlike existing methods, this approach operates on the feature level instead of the pixel level, leading to more efficient OoD detection due to the lower dimensionality of feature encodings. Experiments on multiple established benchmarks, such as CIFAR10 vs. SVHN, LSUN, FMNIST, Omniglot, and CelebA vs. SUN, Textures, and Places365, demonstrate the effectiveness of the OoD detection approach with respect to the evaluation metrics AUROC and FPR95.

**Strengths:**

- Modeling the distribution of feature encodings for OoD detection is an intuitive yet established approach. However, the authors are the first to successfully incorporate generative modeling of features within the framework of OoD detection in image classification tasks.
- The paper is generally well-written and easy to follow. The modules of their OoD detection method—namely, the multi-layer feature extraction, diffusion-based feature reconstruction, and the OoD detection score—are clearly explained. In the ablation studies, the authors demonstrate the effectiveness of each of these modules.
- The proposed method outperforms the state-of-the-art across multiple datasets. In addition to the clear performance gain, the method is also fast. Particularly when compared to other types of generative models, the speedup is significant and relevant in safety-critical applications.

**Weaknesses:**

- The comparisons in the experiments could be extended to ensure fairness. The proposed diffusion and feature-based approach is compared against other types of generative models, such as GLOW and VAE. However, these generative models are only applied at pixel level and rely on image reconstruction error as the OoD score. In my opinion, a fair comparison would involve applying all generative models to the same input, i.e., the multi-scale feature encodings. Different generative models come with their own advantages and drawbacks, and a fair comparison could better show why the diffusion model is the best choice.
- The speed comparison is incomplete. As mentioned, the other generative models are only applied at pixel level, so the time comparison is based on different inputs. Additionally, it would be interesting to see the speed gap between the proposed method and classification or distance-based methods.
- The feature-based approach relies on the feature encodings of the image encoder. From the comparison of EfficientNet-b4 and ResNet50 as encoders, it appears that OoD detection performance benefits from stronger encoders (although this is not explicitly validated). From this perspective, it is unclear what the underlying model is for the classification or distance methods. For a fair comparison, these approaches should also use EfficientNet-b4 and ResNet50 as backbones.
- There are no images to show some qualitative results. Particularly, examples of failure cases would be interesting to see.

**Questions:**

- In line 142ff: “Such multi-layer features could better rebuild the samples and encourage the ID semantic features distributed more compactly within a certain space from different semantic layers.” To me it is unclear what is meant by semantic features being encouraged to be distributed more compactly. As far as I understand, the features of the encoders are only extracted and not modified during the training of the OoD detection model.
- Another insightful metric to measure OoD detection performance could be the AUPRC, in particular if there are significant less OoD examples than ID examples. The numbers for AUROC seem saturated for couple of datasets.
- The OoD detection task with the given datasets seems rather simple. It would be interesting to see the proposed method in other settings, when the original task is more challenging. Are there limitations in terms scalability?

**Limitations:**

The authors have discussed limitations. Potential negative social impact have not been mentioned.

---

> ### Author Rebuttal · Authors · 2024-08-06
>
> Q1. Comparison against other methods using the multi-scale feature encodings as the input.
>
> A: We have made comparison of our method against AE and VAE using the multi-layer feature encodings as inputs.
>
> For AE (AutoEncoder), we use the LFDN network without the timestep embedding, i.e., a 16-layer linear network. For VAE, we use a 5-layer linear network as the encoder and an 8-layer linear network as the decoder.
>
> |ID||OOD|AE(+MFsim)|VAE(+MFsim)|Diffusion(+MFsim)|
> |-|-|-|-|-|-|
> |CIFRA10||SVHN|57.68|83.96|98.89|
> |||LSUN|81.47|97.69|99.83|
> |||MNIST|95.85|99.98|99.99|
> |||FMNIST|79.61|98.69|99.99|
> |||KMNIST|90.51|99.96|99.99|
> |||Omniglot|81.50|97.69|99.99|
> |||NotMNIST|81.61|99.88|99.99|
> |average|||81.18|96.84|**99.81**|
> |Time|Num img/s (↑)||1224.2|1179.4|999.3|
>
> Compared to AE and VAE, the diffusion model has significant advantages when modeling complex multidimensional distributions.
>
> Q2. Speed comparison with classification or distance-based methods.
>
> A: The speed comparison between classification-based and distance-based methods is presented below. All experiments were conducted on an NVIDIA Geforce 4090 GPU.
>
> |Method|MSP|EBO|DICE|ASH-S|SimCLR+Mahalanobis|SimCLR+KNN|ours(+MSE)|ours(+LR)|ours(+MFsim)|
> |-|-|-|-|-|-|-|-|-|-|
> |Type|Classifier-based|Classifier-based|Classifier-based|Classifier-based|Distance-based|Distance-based|Genetive-based|Genetive-based|Genetive-based|
> |img/s (↑)|1060.5|1060.5|1066.3|1047.6|674.8|919.8|960.6|360.2|960.6|
>
> The inference speed of our method based on MSE or MFsim is faster than that of distance-based methods SimCLR+Maha and SimCLR+KNN, because the computation of covariance matrix or K nearest neighbors occupies part of time. Our method is also comparable to classifier-based methods including MSP, EBO, DICE and ASH-S.
>
> Q3. Comparison using the same image encoder as backbone.
>
> A: The experimental results based on classification and distance methods are taken from the best results in the original papers using their optimal backbones.
>
> We also conducted experiments to compare our method with other methods using the same image encoder. The FPR95  and AUROC metrics for methods using EfficientNet-b4 as the backbone are as follows.
>
> We provide the average FPR95 and AUROC values across six OOD datasets: SVHN, LSUN-c, LSUN-r, iSUN, Textures, and Places365.
>
> |ID|Type|Method|**average FPR95**|**average AUROC**|
> |-|-|-|-|-|
> |CIFAR10|Classifier-based|MSP|34.70|94.74|
> |||EBO|17.42|96.31|
> |||DICE|11.55|97.44|
> |||ASH-S|12.53|96.51|
> ||Distance-based|SimCLR+Mahalanobis|57.52|78.87|
> |||SimCLR+KNN|51.53|92.67|
> ||Genetive-based|ours(+MSE)|20.75±0.02|96.93±0.01|
> |||ours(+LR)|17.20±0.02|97.61±0.02|
> |||ours(+MFsim)|**2.51±0.01**|**99.34±0.01**|
> |CIFAR100|Classifier-based|MSP|69.02|80.70|
> |||EBO|84.36|79.31|
> |||DICE|60.39|83.72|
> |||ASH-S|51.71|83.13|
> ||Distance-based|SimCLR+Mahalanobis|89.75|62.71|
> |||SimCLR+KNN|91.56|61.94|
> ||Genetive-based|ours(+MSE)|52.95±0.02|86.35±0.01|
> |||ours(+LR)|51.72±0.03|89.17±0.01|
> |||ours(+MFsim)|**14.78±0.02**|**97.20±0.01**|
>
> The average FPR95 and AUROC metrics of methods using ResNet-50 as the backbone are provided in Table II of the PDF file attached in the global rebuttal.
>
> Q4. Qualitative results.
>
> A: We have included three types of failure cases in the PDF.
>
> The first type, shown in Figure B, represents ID samples misclassified as OOD. It can be observed that these misclassified samples often have significant shadows and lack semantic information, resulting in high reconstruction errors and being incorrectly classified as OOD samples.
>
> The second type, shown in Figure C, represents OOD samples misclassified as ID. It can be observed that these OOD samples have categories very similar to those of the ID samples (CIFAR-10), such as cars and ships, which are categories present in CIFAR-10.
>
> The third type, shown in Figure D, represents OOD samples with colors very similar to the ID samples, leading to their misclassification as ID.
>
> Q5. Explanation to more compact semantic features.
>
> A: Due to the higher dimensionality and complexity of the feature distribution at the pixel level, our proposed multi-scale feature fusion and compression strategy significantly reduces the feature dimensions and makes the distribution more compact. This allows the sample reconstruction process to focus more on the primary features with inter-class discriminative power, rather than secondary details, thereby achieving the goal of distinguishing between ID and OOD samples.
>
> Q6. AUPRC Measurement
>
> A:  We use the AUPRC metric to evaluate OOD detection methods based on the EfficientNet-b4 backbone model. We provide the average AUPRC values across six different datasets in Table I of the PDF. Our method achieves better performance than existing methods as well.
>
> Q7. Experiments on more challenging datasets.
>
> We provide experiments on more challenging datasets using EfficientNet-b4 as the encoder.
>
> (1) We use CIFAR-10 as in-distribution data and CIFAR-100 as OOD data, which has relatively high similarity with CIFAR-10. The results are shown in Table III of the PDF.
>
> (2) We also test OOD detection methods in experiments where ImageNet100 is regarded as the ID dataset and SUN, iNaturalist, Textures, and Places365 are used as OOD datasets. The results are shown in Table IV of the PDF.
>
> Our method still shows significant advantages over the baselines, which demonstrates the scalability of our approach.
>
> Q8. Potential negative social impact have not been mentioned.
>
> A: We have discussed potential negative social impact in the appendix of our paper: As with any advanced detection method, there is a risk that the technology could be misused. For instance, surveillance applications, it could be employed to monitor individuals without their consent, leading to privacy violations and ethical concerns.

---

> > ### Comment · Reviewer_aPqn · 2024-08-08
> >
> > Thank you for the detailed response. I have raised my score accordingly.

---

### Official Review · Reviewer_ch15 · 2024-07-11

**Soundness:** 3
**Presentation:** 3
**Contribution:** 3
**Rating:** 7
**Confidence:** 4

**Summary:**

The paper addresses the problem identification of out of distribution models in an unsupervised manner. The idea is that OOD samples have the largest reconstruction error. The method is evaluated on a plethora of setups including SVHN [Netzer et al., 2011], LSUN(and variants) iNaturalist, Textures, Places365.

**Strengths:**

- The theme is interesting. Identification of the OOD samples has many applications. Limiting factor is that the paper tests in general and synthetic setups for OOD and it does identifies a particular scenarios with OOD samples. Yet compared to other papers which focuses on synthetic evaluation, the setup in this paper is realistic.
- The paper contains innovation, although is somehow limited. To my best knowledge, which is somehow similar with prior work as presented in the paper, the idea to use reconstruction error as a measure of how OOD is a sample,  is indeed novel. Limiting factors lies for instance in triplet loss, where the network is forced to do opposite: adjust weights such that samples which initially were OOD, are pushed towards the middle of the distribution. Yet turning a weak point of a training algorithm in a strong point of a different problems is, in my view, interesting and novel.
- Evaluation is strong: comparison is with strong previous works and in many setups. The evaluation is carried benchmark derivied from 10 datasets and compared against 7 strong solutions. In my view, it is far beyond minimum evaluation required to prove that a method is working.
- ablation is informative. I particularly appreciate the experiment "before and after training"

**Weaknesses:**

- at the core, the idea is that reconstruction error points to OOD. This is new, but is in a broader family (including AE, contrastive leraning), thus some limited
- the limitation sections points only to the strength of the encoder. However the assumption that OOD samples lead to large reconstruction error is also a prior assumption (although a strong one given the evaluation), but one may identify cases where is not true.

**Questions:**

None. Questions I had (e.g. other models as encoder) have been answered in extended material.

**Limitations:**

not the case

---

> ### Author Rebuttal · Authors · 2024-08-06
>
> Q1.at the core, the idea is that reconstruction error points to OOD. This is new, but is in a broader family (including AE, contrastive leraning), thus some limited.
>
> A: Thanks for the insightful comment. We agree that the core idea of using reconstruction error to indicate OOD is indeed related to broader methodologies, including Autoencoders (AEs) and contrastive learning. However, we believe our approach introduces a novel perspective and significant innovations in this field. Our overall framework is innovative in that it applies diffusion models to multi-scale feature layers, establishing reconstruction error from this unique vantage point. This new approach attempts to leverage the strengths of diffusion models in capturing and reconstructing complex data distributions, setting it apart from traditional AE and contrastive learning methods. We appreciate the recognition of the novelty within the broader context and are confident that our contribution offers valuable advancements to the OOD detection domain.
>
>   Additionally, we compared the performance of AE and Diffusion for multi-scale feature modeling to observe their performance differences. For AE (AutoEncoder), we use the LFDN network without the timestep embedding, i.e., a 16-layer linear network. The results are as follows:
>
> |ID|OOD|AE(+MFsim)|Diffusion(+MFsim)|
> |-|-|-|-|
> |CIFRA10|SVHN|57.68|98.89|
> ||LSUN|81.47|99.83|
> ||MNIST|95.85|99.99|
> ||FMNIST|79.61|99.99|
> ||KMNIST|90.51|99.99|
> ||Omniglot|81.50|99.99|
> ||NotMNIST|81.61|99.99|
> ||average|81.18|**99.81**|
> |Time|Num img/s (↑)|1224.2|999.3||
>
> This further demonstrates the advantage of using diffusion for feature modeling.
>
> Q2.the limitation sections points only to the strength of the encoder. However the assumption that OOD samples lead to large reconstruction error is also a prior assumption (although a strong one given the evaluation), but one may identify cases where is not true.
>
> A: Thanks for the valuable feedback. We acknowledge the point regarding the prior assumption that OOD samples lead to large reconstruction errors. While this is generally a prior  assumption supported by our evaluation, there are indeed scenarios where it may not hold true. Some difficult in-distribution (ID) samples may exhibit large reconstruction errors, and certain OOD samples similar to ID samples may have smaller reconstruction errors. The reconstruction errors for ID and OOD samples after training, as shown in Figure 3, also indicate this point. There is a small overlap between the reconstruction errors of some ID and OOD samples.
>
> We also considered the detection performance of our method when the reconstruction errors are quite close. For example, the ID dataset is from CIFAR10, and the OOD dataset is CIFAR100 which shares similar distributions with CIFAR10. Below are our results:
>
> CIFAR-10 as ID and CIFAR-100 as OOD:
>
> |ID|OOD|Method|FPR95↓|AUROC↑|
> |-|-|-|-|-|
> |CIFAR10|CIFAR100|MSP|52.04|86.14|
> |||EBO|51.32|86.19|
> |||ASH-S|51.29|87.13|
> |||GLOW|➖|73.60|
> |||VAE|90.41|55.95|
> |||DDPM|93.21|54.00|
> |||ours(+MSE)|**48.87**|**87.54**|
> |||ours(+LR)|49.48|87.24|
> |||ours(+MFsim)|53.70|85.60|
>
> CIFAR-10 as ID and noisy CIFAR-10 as OOD:
>
> |ID|OOD|Method|FPR95↓|AUROC↑|
> |-|-|-|-|-|
> |CIFAR10|CIFAR10(add noise)|MSP|90.23|61.63|
> |||EBO|88.01|62.49|
> |||ASH-S|86.95|62.16|
> |||VAE|89.20|51.76|
> |||DDPM|95.80|45.00|
> |||ours(+MSE)|7.84|98.43|
> |||ours(+LR)|16.27|96.76|
> |||ours(+MFsim)|**0.14**|**99.97**|
>
> It can be observed that our method still achieves the best performance in these two specific scenarios. This validates the reasonableness of this prior assumption.

---

> > ### Comment · Reviewer_ch15 · 2024-08-12
> >
> > Thank you for your thoughtful and detailed answer!

---

### Official Review · Reviewer_yKR8 · 2024-07-17

**Soundness:** 3
**Presentation:** 2
**Contribution:** 2
**Rating:** 6
**Confidence:** 4

**Summary:**

This paper proposes a diffusion-based layer-wise semantic reconstruction strategy for unsupervised Out of Domain (OOD) detection. Specifically, they leverage the intrinsic data reconstruction ability of the diffusion model to differentiate between the In-distribution (ID) and OOD samples in the feature space. The features/data reconstruction at the multi-layer feature spaces helps the generative OOD detection.  The experiments suggest the superiority of the proposed approach compared to existing approaches.

**Strengths:**

- The paper proposes a novel scheme for unsupervised OOD detection based on semantic features  reconstruction at multiple layers.

- Building the diffusion network on top of multi-level features instead of pixel-level outputs, helps in better preserving the ID information.

- State-of-the-art performance on benchmark datasets.

**Weaknesses:**

1- The paper needs a writeup revision, for example the contributions and idea of the paper is repeated multiple times unnecessarily.

2- The third step of the proposed approach, I.e., OOD detection step, is not clear. The authors defines three metrics for OOD detection based on some threshold, however, the paper never mentioned those values nor provided any experimental  study, that how were they selected.

3- The comparison with recent generative methods based OOD is not provided. Instead only VAE is results are compared, despite there are some approaches available ([Graham et al., 2023], [Gao et al., 2023] and [Liu  et al., 20]).

4- A comparative analysis between the pixel level and feature level denoising is not performed, which is the base of the proposed approach.

5-The authors needs to clearly indicate that how the proposed approach is better than the existing pixel level approaches, both theoretically and experimentally.

6- The proposed approach, and most of the methods cited in this paper experimented with classification problems. However, they build  their motivation and need based on the natural scenarios. In natural scenarios, the input images may be more complex and the task may be more difficult than classification only. A study is required to evaluate the proposed approach on complex scenes, object detection and semantic segmentation datasets, having more information instead of having a single primary object.

**Questions:**

The questions are there in the weaknesses section, specifically, experimental evidence of points 2-6 in weaknesses section will significantly improve the completeness.

**Limitations:**

The authors mentioned about the limitations related to backbone architecture. However, the large models pre-trained on large datasets are more generalised. So, how this generalisation will effect the models performance?

---

> ### Author Rebuttal · Authors · 2024-08-07
>
> Q1. Writeup revision.
>
> A: Thank you for your valuable feedback. We appreciate your observation regarding the repetition of the contributions and the main idea of the paper. We will address these issues to ensure clarity and conciseness.
>
> Q2. Clarification of OOD detection step.
>
> A: During inference, samples having higher reconstruction errors measured by MSE, MFsim, or LR  are more likely to be OOD samples.  FPR95 indicates the false positive rate when the true positive rate reaches 95\%. When calculating FPR95, the threshold is determined by the measurement value where the true positive rate (TPR) reaches 95%. AUROC measures the area under the ROC curve. When calculating this metric, we vary the true positive rate from xxx to xxx in step of xxx to choose the threshold. In practical usage, we can select the threshold value with the OTSU algorithm. We report the F-scores of different methods using OTSU to determine the threshold value as follows:
>
> |ID|Method|SVHN|LSUN-c|LSUN-r|iSUN|Textures|Places365|**average**|
> |-|-|-|-|-|-|-|-|-|
> |CIFAR10|MSP|48.20|80.88|75.74|75.85|81.75|75.90|73.05|
> ||EBO|46.76|92.72|81.93|81.30|82.53|81.17|77.74|
> ||ASH-S|67.44|93.55|90.99|91.33|83.12|82.39|84.80|
> ||ours(+MSE)|79.21|91.04|83.16|82.93|97.43|97.43|88.53|
> ||ours(+LR)|86.18|91.85|85.74|85.03|97.43|97.43|90.61|
> ||ours(+MFsim)|91.91|97.42|95.20|94.71|97.43|97.43|**95.68**|
> |CIFAR100|MSP|47.92|72.47|70.29|72.08|78.51|68.02|68.22|
> ||EBO|46.81|75.34|70.91|72.39|78.48|67.84|68.63|
> ||ASH-S|44.35|74.83|67.94|69.86|78.20|68.79|67.33|
> ||ours(+MSE)|49.80|73.75|67.56|70.11|97.43|97.43|76.01|
> ||ours(+LR)|52.39|74.05|68.75|71.13|97.43|97.42|76.86|
> ||ours(+MFsim)|65.48|96.52|87.39|87.20|97.43|97.43|**88.58**|
>
> Q3. Comparisons with recent generative methods based OOD
>
> A:  The comparison between our method and DDPM [Graham et al., 2023] can be referred to Table 1 and 2 of our paper.  Our method outperforms DDPM consistently on benchmarks using CIFAR10 or CelebA as ID data.
>
> The comparison between our method and Diffuard [Gao et al., 2023] is provided in the table below. Results of Diffuard are taken from its original paper. Here, CIFAR10 is regarded as ID data, while CIFAR100 or TinyImagenet is regarded as OOD data. Our method based on MFsim achieves overall better performance than ‘Diffuard+Deep Ens’, with 1.55 higher AUROC and 21.77 lower FPR95.
>
> |Method|CIFAR-100 AUROC↑|CIFAR-100 FPR95↓|TINYIMAGENET AUROC↑|TINYIMAGENET FPR95↓|average AUROC↑|average FPR95↓|
> |-|-|-|-|-|-|-|
> |Diffuard|89.88|52.67|91.88|45.48|90.88|49.08|
> |Diffuard+EBO|89.93|50.77|91.95|43.58|90.94|47.18|
> |Diffuaed+Deep Ens|**90.40**|52.51|91.98|45.04|91.19|48.78|
> |ours(+MSE)|87.54|**48.87**|97.68|13.42|92.61|31.15|
> |ours(+LR)|87.24|49.48|97.11|15.04|92.18|32.26|
> |ours(+MFsim)|85.6|53.7|**99.88**|**0.39**|**92.74**|**27.01**|
>
> The comparison between our method and LMD [Liu et al., 2023] is shown in the following table. The evaluation metric is AUROC. The average AUROC of our method based on MFsim is 6.94 higher than that of LMD.
>
> |ID|OOD|LMD|ours(+MSE)|ours(+LR)|ours(+MFsim)|
> |-|-|-|-|-|-|
> |CIFAR10|CIFAR100|60.70|87.54|87.24|85.6|
> ||SVHN|99.20|97.31|98.22|98.89|
> |CIFAR100|CIFAR10|56.80|70.52|72.86|64.58|
> ||SVHN|98.50|83.93|88.84|93.9|
> ||**AVERAGE**|78.80|84.83|**86.79**|85.74|
>
> Q4. Comparisons with pixel-level denoising approaches.
>
> A: The quantitative comparisons between our method and pixel-level approaches are as follows:
>
> We provide the distribution differences of the MSE scores at two levels after training, with CIFAR-10 as the ID dataset and other datasets as OOD; The results are shown in Figure A of the PDF attached in the global rebuttal.
>
> It can be observed that at the pixel level, the reconstruction error distributions of ID and OOD samples are very similar. The mixed MSE scores make it very hard to distinguish ID samples from OOD samples. However, at the feature level, the reconstruction score distribution of ID samples shows a clear distinction from that of OOD samples. The reason is that, our feature-level diffusion-based generative model makes the projected in-distribution latent space not only be compressed sufficiently to capture the exclusive characteristics of ID images, but also provide sufficient reconstruction power for the large-scale ID images of various categories.
>
> Q5. Focusing on Classification Problems
>
> A: Thanks for the valuable comment. Our experiments, along with most of the cited works, focus on classification problems. This focus is largely because the field of OOD detection has traditionally concentrated on classification tasks. However, we believe our method has the potential to extend beyond classification. Our approach can serve as an effective data pre-processing step for more complex tasks like semantic segmentation and object detection. By identifying and filtering out OOD inputs, our method can enhance data security and improve the performance of these models. Additionally, our method can be integrated into semantic segmentation and object detection pipelines to judge whether detected or segmented object proposals are out of distribution.
>
> Q6. Effect of generalisation on model performance.
>
> A generalised backbone model is crucial for ensuring the extracted features are comprehensive and representative of the samples. When modeling in multi-scale feature spaces, it's essential that the features extracted are thorough and meaningful. Small-scale models may have shallower layers, which might not fully capture the complexity of a sample. The generalization ability of large models allows them to extract a more complete and representative set of features, which is essential for the effective performance of our OOD detection method. This ensures that the model can adequately generalize across various inputs, leading to a more robust and reliable detection system.

---

> > ### Comment · Reviewer_yKR8 · 2024-08-08
> >
> > Thank you for the detailed response.

---

> > ### Comment · Reviewer_yKR8 · 2024-08-12
> > **Queries addressed**
> >
> > I have revised my rating accordingly.

---

### Author Rebuttal · Authors · 2024-08-06

We thank all reviewers for their valuable feedback, with three reviewers (yKR8, ch15 and aPqn) supporting our work. We are encouraged that reviews think our paper:
- **A novel and interesting unsupervised OOD detection scheme.** (by Reviewer yKR8, ch15, aPqn)
- **State-of-the-art performances on benchmark datasets.** (by Reviewer yKR8, ch15, aPqn)
- **Strong evaluation and informative ablation.** (by Reviewer ch15)
- **Good presentation.** (by Reviewer aPqn, Jgzd)


Concerns of reviewers are addressed in the rebuttal to each reviewer with extra tables and figures in the attached pdf document.

1.In response to Reviewer aPqn’s question on evaluation metric, we provide AUPRC to measure OOD detection performance in Table I.

2.In response to Reviewer aPqn’s concern on using same backbone, we provide the average FPR95 and AUROC values for different methods using ResNet-50 as the backbone in Table II.

3.In response to Reviewer aPqn’s concern on scalability to more challenging settings, we conduct experiments using CIFAR-10 as the ID dataset and CIFAR-100 as the OOD dataset. The results are presented in Table III. We also conducted experiments using ImageNet100 as ID dataset. The results are presented in Table IV.

4.In response to Reviewer yKR8’s question on comparative analysis between the pixel-level and feature-level denoising, we visualize the reconstruction error distribution for both pixel-level and feature-level reconstructions in Figure A. It can be observed that at the pixel level, the reconstruction error distributions of ID and OOD samples are very similar. However, at the feature level, the reconstruction score distribution of ID samples shows a clear distinction from that of OOD samples.

5.In response to Reviewer aPqn’s request on qualitative results of failure cases, we provide three groups of samples representing three main types of failure cases.

We have tried our best to answer all the reviewers’ questions about our paper. We sincerely hope that our responses can address all the concerns.

---

### Decision · Program_Chairs · 2024-09-25

**Decision:**

Accept (poster)

**Comment:**

Dear authors,

This draft has received 1 Accept, 2 Weak Accept, and 1 borderline. After carefully reviewing the comments, the draft is being accepted for publication. Authors are encouraged to update the draft according to the suggestions and concerns mentioned by the reviewers.

Congratulations.

regards
AC